# A live attenuated vaccine to prevent severe neonatal *Escherichia coli* K1 infections

Youssouf Sereme ®[1], Cécile Schrimp[1], Helène Faury[1,2], Maeva Agapoff ®[1], Esther Lefebvre-Wloszczowski[1], Yunhua Chang Marchand[1], Elisabeth Ageron-Ardila[1], Emilie Panafieu[3], Frank Blec[3], Mathieu Coureuil[1], Eric Frapy[1], Vassilis Tsatsaris[4,5], Stephane Bonacorsi[6,7] & David Skurnik ®[1,2,5] ✉

Preterm birth is currently the leading cause of neonatal morbidity and mortality. Genetic, immunological and infectious causes are suspected. Preterm infants have a higher risk of severe bacterial neonatal infections, most of which are caused by *Escherichia coli* an in particular *E. coli* K1strains. Women with history of preterm delivery have a high risk of recurrence and therefore constitute a target population for the development of vaccine against *E. coli* neonatal infections. Here, we characterize the immunological, microbiological and protective properties of a live attenuated vaccine candidate in adult female mice and their pups against after a challenge by K1 and non-K1 strains of *E. coli*. Our results show that the *E. coli* K1 E11 Δ*aroA* vaccine induces strong immunity, driven by polyclonal bactericidal antibodies. In our model of meningitis, mothers immunized prior to mating transfer maternal antibodies to pups, which protect newborn mice against various K1 and non-K1 strains of *E. coli*. Given the very high mortality rate and the neurological sequalae associated with neonatal *E. coli* K1 meningitis, our results constitute preclinical proof of concept for the development of a live attenuated vaccine against severe *E. coli* infections in women at risk of preterm delivery.

Preterm delivery (<37 weeks of gestation) is the leading cause of neonatal morbidity and mortality[1]. Each year, approximately 15 million babies worldwide are born prematurely; this corresponds to 11% of all births. 90% of preterm deliveries take place in low-income/resource-poor countries[2,3]. With 35% of all deaths among newborns due to prematurity, prematurity which also accounts for 18% of all deaths among children aged under 5 years is a true public health problem worldwide[2,3]. The etiology of preterm delivery is multifactorial, and genetic, infectious and immunological factors have been described[4,5]. Researchers have found that women with a previous preterm delivery are significantly more likely to have a subsequent preterm delivery[5]. Furthermore, infections are at least six times more frequent in preterm

infants than in term infants[6], due to immaturity of the immune system[7]. Indeed, preterm infants have functional impairments in innate immunity (as evidenced by abnormally low levels of cytokines, chemokines, antimicrobial proteins, and antimicrobial peptides), cellular immunity, and the complement system[7–10]. *Escherichia coli* an in particular E. coli K1 strains among the most common pathogens in newborns and constitutes the leading cause of neonatal bacterial meningitis in preterm infants[11–13]. Infants born to women with a previous preterm delivery have an elevated risk of neonatal *E. coli* K1 meningitis; hence, vaccine-based prevention in these women might significantly reduce the incidence of neonatal meningitis. Scientific and technological progress has enabled the development of a vaccine

[1]Université Paris Cité, CNRS, INSERM, Institut Necker Enfants Malades, Paris, France. [2]Department of Microbiology, Necker Hospital, University de Paris, Paris, France. [3]LEAT antenne Imagine- SFR Necker INSERM US 24, Paris, France. [4]Maternité Port-Royal, hôpital Cochin, GHU Centre Paris cité, AP-HP, Paris, France. [5]FHU PREMA, Maternité Port-Royal, Paris, France. [6]IAME, UMR 1137, INSERM, Université Paris Cité, Paris, France. [7]Laboratoire de Microbiologie, Hôpital Robert Debré, AP-HP, Paris, France. ✉e-mail: david.skurnik@inserm.fr

against *E. coli* K1 bacterial meningitis. Although a candidate vaccine against *E. coli* K1 (based on the OmpA protein) has shown efficacy in mice[14–16], it has not been approved to date for use in clinical practice.

Live attenuated vaccines reportedly confer the high levels of rapid, sustained, full immunity required for protection of the mother and her infant[17,18]. However, the use of these vaccines is not recommended in certain high-risk populations, such as pregnant women and immunocompromised individuals[19,20]. Nevertheless, the administration of a live vaccine is not contraindicated prior to pregnancy in women of childbearing age or in those at risk of preterm delivery[20–22].

By using a combination of saturated transposon mutagenesis and high-throughput sequencing (TnSeq, a powerful tool to study host-pathogen interactions), we recently identified the *aroA* gene as a virulence factor in *E. coli* K1[23]. *aroA* encodes 5-enolpyruvylshikimate-3-phosphate synthase which is involved in the biosynthesis of aromatic amino acids[24] and several iron capture systems[25]. Therefore, we decided to build a *ΔaroA E. coli* K1 mutant and evaluate it potential as a live attenuated vaccine against severe neonatal infections caused by *E. coli*. Although our TnSeq experiments had revealed other *E. coli* K1 virulence factors, AroA was selected for live vaccine attenuation because deletion of the *aroA* gene have been shown to provide strong protection in other settings (such as *Pseudomonas aeruginosa* or *Salmonella* sp. Infections)[26,27]. Furthermore, it has also been reported that an *E. coli* O78 vaccine attenuated by deletion of the *aroA* gene was efficacious against avian diseases[28]. One of the main reason *aroA* is a good mutated background for vaccine strains is that it makes strains auxotrophic for aromatic amino acids and so cannot replicate properly in certain niches in the host[26,29,30].

In the present study, we first confirmed the results of the Tnseq data experiments and showed that the virulence of *E. coli* K1 E11 *ΔaroA* was indeed attenuated both in vitro and in vivo probably through the significant decreased of the expression of the genes encoding for type 1 fimbriae, a major *E. coli* K1 virulence factor[31,32] in the *ΔaroA* mutant strains. We then characterized the immunological and microbiological properties of a live attenuated *E. coli* K1 E11 *ΔaroA* vaccine candidate after injection into adult female mice. Lastly, we demonstrated the very strong degree of protection afforded to offspring born to mothers having received with the attenuated vaccine before mating. Our results constitute preclinical proof of concept of a new strategy for the development of a vaccine against *E. coli* K1 infections in women at risk of preterm delivery.

## Results

### Characterization of the attenuation of E. coli K1 E11ΔaroA

Using the methods described by Datsenko[33], we built a *ΔaroA* mutant by deleting the gene *aroA* from a clinical strain of *E. coli* K1 E11[23]. The absence of the *aroA* gene in the mutant strain was confirmed by a PCR analysis and control experiments. Clones resistant to 100 μg/mL kanamycin were obtained and then tested (using PCR) to check the insertion of the kanamycin cassette and the deletion of the *aroA* gene (Supplementary Fig. 1).

After seven hours of culture in liquid lysogeny broth (LB) and Dulbecco's Modified Eagle Medium (DMEM) medium, the growth rate of the *aroA*-deleted strain was similar to that of the wild type (WT) *E. coli* E11 strain in LB medium, While a non-significant reduction ($P > 0.05$) was found in DMEM, the culture medium used in the cell experiments conducted in this project (Fig. 1A) when carrying out an auxotrophicity test for amino acids in the minimal M9 medium, we observed a difference in growth between *E. coli* K1 E11 WT and *ΔaroA* in M9 minimal medium not supplemented with amino acid: the mutant strain was not able to grow in minimal medium unless supplemented with amino acids (Supplementary Fig. 1B).

Then, we found that the adhesion and invasion capacities of *E. coli* K1 E11 *ΔaroA* were significantly reduced compared to those of the WT strain ($p = 0.002$ and $p = 0.02$ respectively) (Fig. 1B, C) without any bias

that could have been caused by a growth defect of the *AroA* mutant (no difference between the two strains at T0 of the experiment as shown in Supplementary Fig. 1C–E).

While part of this finding could be due to the growth defect of the *E. coli* E11 *ΔaroA* strain found in DMEM (Fig. 1A), we decided to further explore the phenotype of the *aroA* mutant, in particular with regard to any changes in expression of the main *E. coli* K1 virulence determinants. Therefore, we performed a comprehensive investigation on the impact on the transcriptome of the deletion of the *aroA* gene by conducting RNAseq experiments in LB and DMEM of *E. coli* E11 WT and *ΔaroA*.

As shown (Fig. 1D) and (Supplemental Table 1), genes encoding for type 1 fimbriae were among the top ten genes with the most decreased expression in the *E. coli* E11 *ΔaroA* compared to E11 WT when both strains were grown in LB. This decreased expression was confirmed by qRT-PCR (Fig. 1E) and also found in DMEM (Supplemental Fig. 5). When looking at the top ten percent genes with the most decreased expression in *E. coli* E11 *ΔaroA* compared to E11 WT in LB and the top ten percent genes with the most decreased expression in *E. coli* E11 *ΔaroA* compared to E11 WT in DMEM, we found 39 genes in common (Supplemental Table 2). Among these 39 genes, we found all the genes encoding for the Type 1 Fimbriae except the regulator genes *fimB* and E[34]. The role of the Type 1 Fimbriae in the adhesion on epithelial cells has long been described in *E. coli*, further explaining the results presented Fig. 1B–C. However, the Type 1 Fimbriae also constitutes a major virulence factor in *E. coli* K1[31,32]. Therefore, we next evaluated the in vivo attenuation of *E. coli* K1 E11*ΔaroA* compared to *E. coli* K1 E11 WT.

Using a mouse model of systemic dissemination after intraperitoneal injection, we compared the mortality rates resulting from infections by the mutant strain *E. coli* K1 E11 *ΔaroA* vs. the WT strain (Fig. 1F). We also tested two *E. coli* K1 strains (*E. coli* S88[35] and RS218[36]) and two non-K1 *E. coli* meningitis strains (*E. coli* S510 and *E. coli* S242) responsible for severe neonatal infections (Fig. 1G, H). All the *E. coli* K1 strains were 100 to 1000 times more virulent than the *E. coli* K1 E11 *ΔaroA* mutant, while the *E. coli* non-K1 strains were 10–100 times more virulent than the *ΔaroA* mutant. Indeed, the lethal doses were $10^7$ colony forming units (CFU) for *E. coli* E11 WT and *E. coli* S88, $10^6$ CFU for *E. coli* RS218, $10^7$ CFU for *E. coli* S510 (non-K1), $10^8$ CFU for *E. coli* S242, and $>10^9$ CFU for the mutant strain *E. coli* E11 *ΔaroA* (Fig. 1F–H). Our data showed that *E. coli* E11 *ΔaroA* could be used safely as a live attenuated vaccine in our project—even at doses that would be lethal if non-mutant strains were used. The significant reduction in the expression of genes coding for type 1 fimbriae found in this study probably add significantly to the safety of the live attenuated *E. coli* E11 *ΔaroA* vaccine.

### Evaluation of the humoral immune response after immunization

After having confirmed the safety of our live attenuated vaccine, we evaluated its immunogenicity in an immunization regimen (Fig. 2A) with intraperitoneal (IP) and/or subcutaneous (SC) injection of a $10^5$ CFU suspension of *E. coli* E11 *ΔaroA* in adult female mice once a week for 3 weeks.

Vaccinated female mice were sampled to test their humoral response against whole-bacterial cells. Levels of specific antibodies against three *E. coli* K1 strains (*E. coli* E11 WT, S88 and RS218) initially selected for the study were detected (using an ELISA) in the sera of immunized mice by IP route (Fig. 2B). Levels of specific antibodies against *E. coli* K1 strains (*E. coli* E11 WT and RS218) were also detected (using ELISA) in the serum of subcutaneously immunized mice (Fig. 2C). The kinetics of antibody titers after immunization with 3 doses by the IP and SC routes were also observed. High antibody levels were detected up to 35 days after immunization (Fig. 2D) and more that 100 days after immunization by IP route (Supplementary Fig. 2A). We also detected isotype levels of IgG1, IgG2a, IgG2b and IgG3 in the serum of immunized mice. The IgG2 level was the highest in the serum

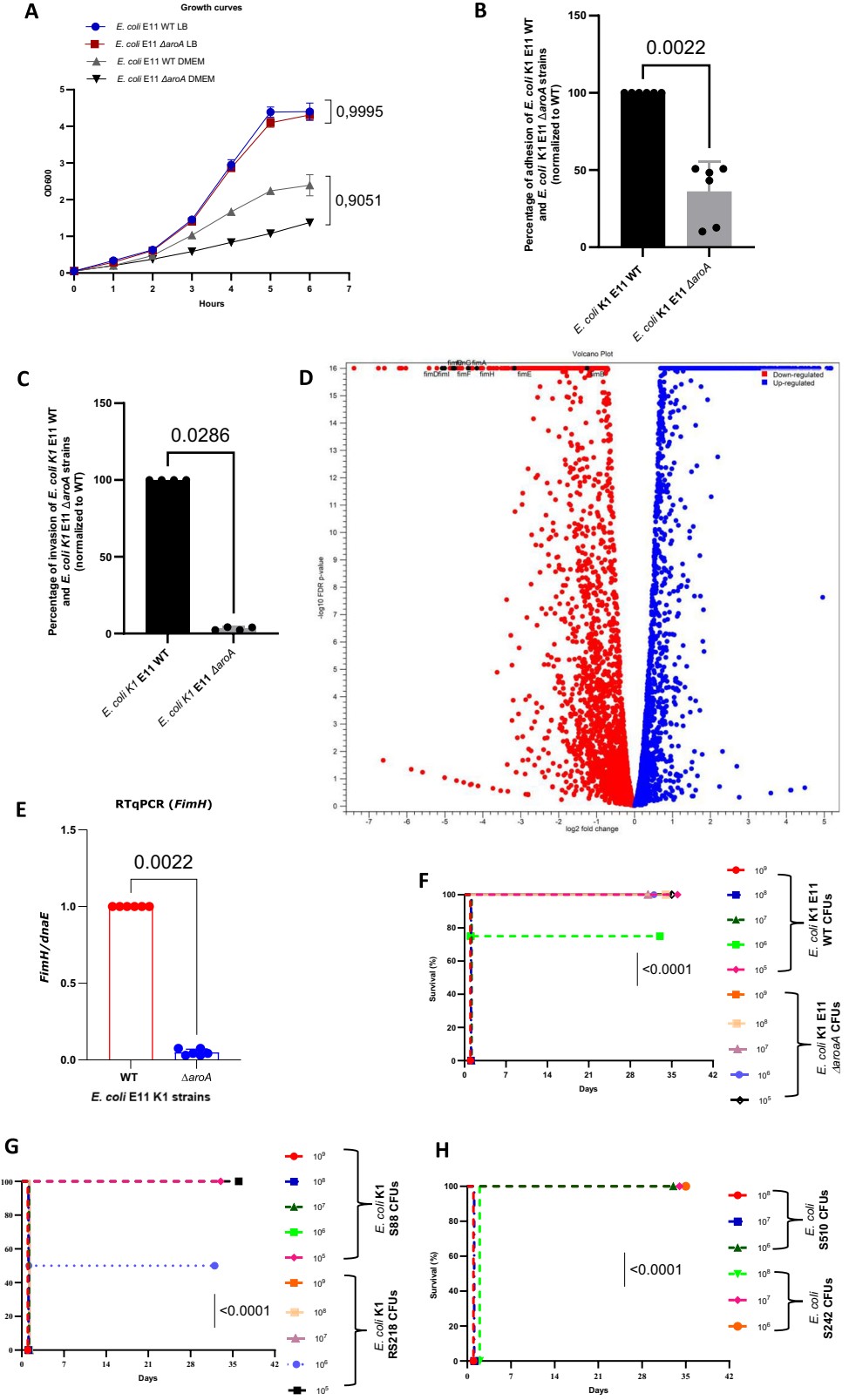

tested (Fig. 2E). In the sera of immunized mice, we also found an elevated titer of antibodies against other *E. coli* K1 and *E. coli* non-K1 strains including strains responsible for severe neonatal infections (provided by the French National Reference Center) (Supplementary Fig. 2B, C). To confirm that our live attenuated vaccine induced a specific, polyclonal immune response, we next performed SDS PAGE gel experiments using *E. coli* K1 strains and Western blot experiments

with three *E. coli* K1 strains and a *E. coli* non-K1 strain. We were able to visualize the proteins in all of these bacteria (Supplementary Fig. 2D) and we detected polyclonal antibodies against several bacterial antigens (Supplementary Fig. 2E, F), including several antigens from the *E. coli* non-K1 S510 strain. (Supplementary Fig. 2G). Furthermore, we compared the immunogenicity of the live attenuated strain to the inactivated strain, and found

**Fig. 1 | Characterization and evaluation of the virulence of the *E. coli* K1 E11 *ΔaroA* strain. A:** Growth curves for *E. coli* E11 WT and *ΔaroA* strains in LB and DMEM medium. After 7 h of culture, the difference in the growth rate between the two strains was not significant ($p > 0.05$) ($n = 3$). **B, C** Adhesion to (**B**) ($n = 6$) and invasion (**C**) ($n = 4$) of *Hela* cells by *E. coli* E11 *ΔaroA* were significantly lower than for the WT strain *E. coli* E11 ($p < 0.05$) data are presented as mean values SD. **D** RNAseq analysis of *E. coli* E11 WT and *E. coli* E11 *ΔaroA* grown in LB: genes encoding for the genes *fimABCDEFGHI* (red dots) are annotated in a volcano plot: a scatterplot showing statistical significance (*P*-values, *Y*-axis) versus magnitude of change (fold change, *X* axis) ($n = 3$). **E** Expression of the *FimH* gene by RTqPCR in the E11 WT vs *ΔaroA* strain in LB. Significantly decreased expression of the *FimH* is found in E11 *ΔaroA* ($p = 0.02$) ($n = 6$). **F−H** Evaluation of the in vivo ($n = 4$) mortality of the mutant strain vs. the virulent WT *E. coli* K1 E11, S88, RS218 (**F-G**) and two non-K1(H) *E. coli* strains. The WT strains were 100 times more virulent than the mutant strain (**F**). The lethal dose was $10^7$ CFU for *E. coli* K1 E11 WT and S88, $10^6$ CFU for RS218, $10^8$ CFU for S242 (no-K1), $10^7$ CFU for S510 (no-K1) and over $10^9$ CFU for the mutant strain. Error bars represent standard deviations (SD). (**A**): *p*-value values indicate significant differences in an ordinary One-way ANOVA, Dunnett's multiple comparisons test, with a single pooled variance. (**B**, **C**, and **E**) *p*-value values indicate significant differences in a two-tailed Mann−Whitney non-parametric test (**B**, **C**, and **E**), (**D**): one-way ANOVA using Bonferroni's multiple comparisons test and a Mantel−Haenszel log-rank test (**F−H**). Source data are provided as a Source Data file.

that the live attenuated strain was significantly more immunogenic compared to the inactivated strain (Supplementary Fig. 2J). Taken as a whole, these results show that immunization with the *ΔaroA* mutant *E. coli* K1 strain induced a strong humoral immune response and triggered the production of polyclonal antibodies capable of binding to a range of antigens from both K1 and non-K1 strains of *E. coli*.

## Evaluation of the cellular immune response after immunization

We next evaluated the cellular immune response in BALB/c female mice ($n = 4$) immunized with three doses of the *E. coli* E11 K1 *ΔaroA* suspension. Mice were sacrificed 7 days after the third dose of immunization; each animal's spleen was collected, and the splenocytes were isolated and analyzed. Our results showed that there were no significant differences between vaccinated and nonvaccinated groups of mice with regard to the expression of T cell (Fig. 3A), B cell (Fig. 3B), CD4 and CD8 T cell populations (Fig. 3C, D) and regulatory T cells (Fig. 3E). However, a trend towards a greater regulatory T cell count was observed in the spleen samples from vaccinated mice but with no significant difference (Fig. 3F, G). Furthermore, we assessed the cellular response at the systemic level by quantifying serum levels of cytokines associated with Th1, Th2 and Th17 responses. The production of TNFα (Fig. 3H) and IL-10 (Fig. 3I) was induced more strongly in the vaccinated mice than in the nonvaccinated mice, even though the levels of these cytokines were very low in absolute terms. Our results indicate that vaccination with the *E. coli* K1 E11 *ΔaroA* strain did not induce a robust cellular immune response.

## Evaluation of the protection of vaccinated female mice

To evaluate the in vitro protection afforded by the polyclonal antibodies generated after vaccination, we performed serum bactericidal assays against three strains of *E. coli* K1 (E11 WT, S88 and RS218). In the presence of baby rabbit complement, we detected strong bactericidal activities against the three *E. coli* K1 strains in the serum of immunized mice, up to a serum dilution of 1:100 (Fig. 4A). We detected bactericidal activity against two other K1 strains of *E. coli* (S524 and S395) in the serum of immunized mice at a serum dilution of 1:2 (Supplementary Fig. 3A). However, the K1 capsule is known for its high resistance to the action of the complement[37] and other strains could be more or totally resistant if tested in this assay[38,39]. To confirm the specificity of the observed bactericidal activity, we tested other pathogenic Gram-negative bacteria (*Klebsiella pneumoniae* and *Pseudomonas aeruginosa* PA14, previously used in killing experiments) as controls. The absence of bactericidal activity against these two control species confirmed the specificity of our assays (Fig. 4B).

To further evaluate the protection conferred by our vaccine candidate against *E. coli* K1 infections, adult female mice immunized by IP route were challenged with two lethal doses ($10^7$ and $10^8$ CFU/mouse) of three *E. coli* K1 strains (E11 WT, S88 and RS218). We also assessed the protection conferred by our vaccine against a *E. coli* non-K1 S510 (lethal dose: $10^7$ CFU/mouse). All the vaccinated mice

challenged with the three *E. coli* K1 strains and *E. coli* non-K1 S510 survived, up to 35 days after the infection. Conversely, we observed a 100% mortality rate within 24 h after nonvaccinated control mice were challenged with the same *E. coli* K1 strains (Fig. 4C−H). Mortality rates of 75% and 100% were observed at 48 and 72 h, respectively, in the nonvaccinated group infected with the *E. coli* non-K1 S510 strain (Fig. 4I). A 100% protection was also observed in adult challenged mice 60 days after immunization with *E. coli* strains K1 E11 WT, S88 and RS218 (Supplementary Fig. 3B−D).

Adult female mice immunized by the SC route were infected with a lethal dose $10^7$ CFU/mouse by three strains of *E. coli* K1 (E11 WT, S88 and RS218). All the vaccinated mice challenged with the three *E. coli* K1 strains survived, up to 35 days after the infection. Conversely, we observed a 100% mortality rate within 24 h after nonvaccinated control mice were challenged with the same *E. coli* K1 strains (Supplementary Fig. 4A−C).

Taken as a whole, these results indicate that vaccination of a mouse model with the attenuated vaccine *E. coli* K1 E11 *ΔaroA* induced a strong polyclonal antibody-mediated humoral response and led to significant protection. In vitro, the vaccine-induced antibodies mediated the killing of *E. coli* K1. In vivo, our vaccine candidate protected adult mice against severe infections caused by *E. coli* K1 (E11 WT, S88 and RS218) and *E. coli* non-K1 (S510).

## Evaluation of the presence of maternal antibodies in mouse pups

To assess the vertical transfer of maternal antibodies to mouse pups, female mice were immunized with three doses of our vaccine and then mated with male mice with the same BALB/c genetic background (Fig. 5A). The resulting pups were sacrificed at the age of 7 days. Blood samples were collected, and titers of *E. coli* K1-specific antibody were quantified in a whole-bacterial cell ELISA. We found antibodies able to bind to several strains of *E. coli* K1 in the sera of pups born to immunized mothers but not in the sera of pups born to nonvaccinated mothers (Fig. 5B). Western blot experiments confirmed that the maternal antibodies found in the sera of pups born to immunized mothers were specific and polyclonal (Supplementary Fig. 2H). Conversely, no specific, polyclonal antibodies were found in the serum of pups born to nonvaccinated mothers (Supplementary Fig. 2I). Taken as a whole, these results evidenced the vertical transfer of maternal antibodies to the pups.

## The effect of maternal vaccination on the protection of the pups

The protection of pups born to vaccinated mothers by IP and SC route was evaluated at three and 7 days of life.

Three-day-old pups born to mothers immunized by IP were challenged with $10^5$ CFU of *E. coli* K1 S88 (Fig. 6A), $10^5$ CFU of *E. coli* RS218 (Fig. 6B), and $10^5$ CFU of *E. coli* E11(Fig. 6C). Three-day-old pups from mothers immunized by SC were challenged with $10^5$ CFU of *E. coli* K1 S88, $10^5$ CFU of *E. coli* RS218, and $10^5$ CFU of *E. coli* E11(Supplementary Fig. 4D−F) Very strong protection was observed against all

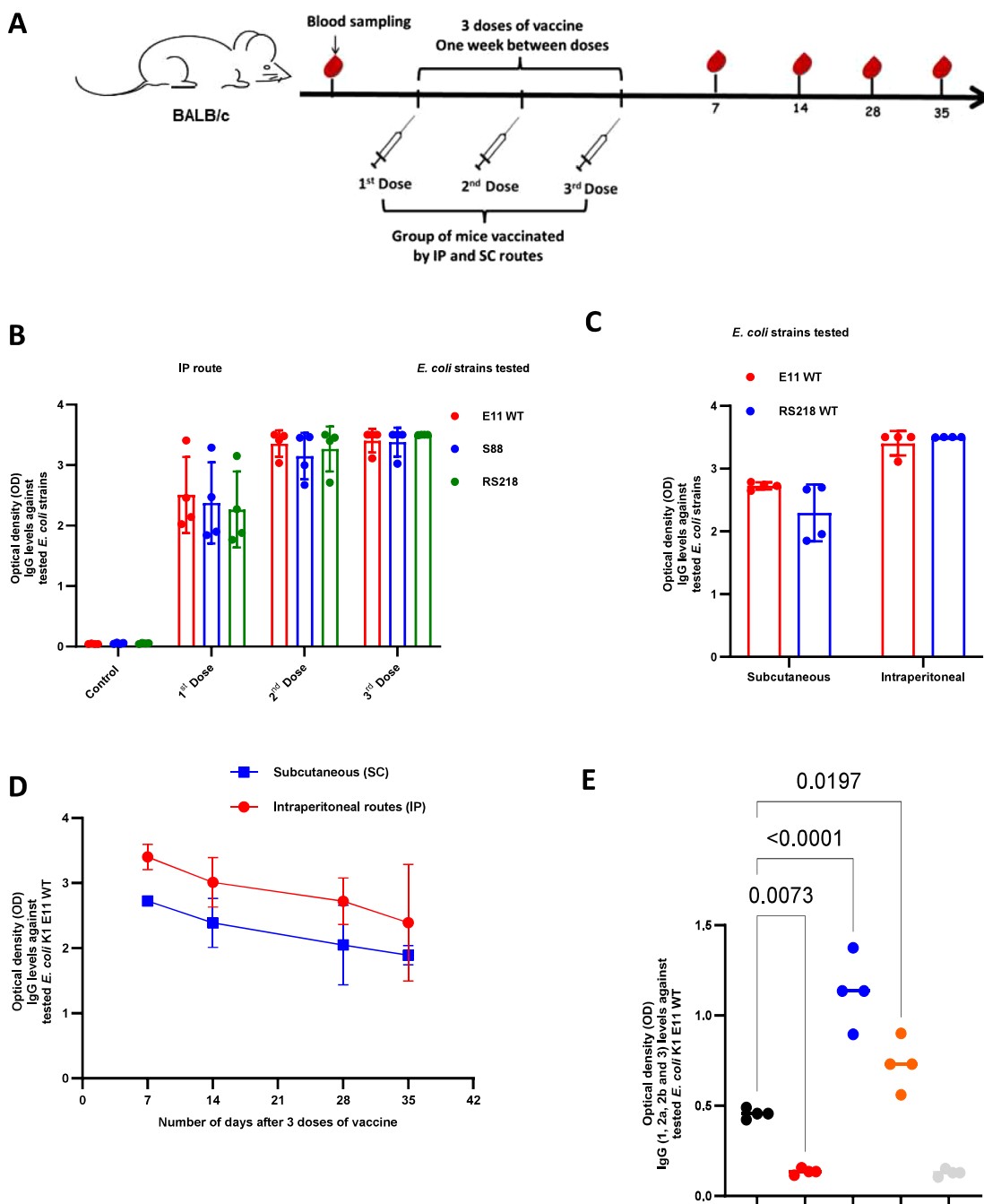

**Fig. 2 | Evaluation of the humoral response after immunization with *E. coli* K1 E11 Δ*aroA*. A** The immunization regimen for BALB/c female mice with $10^5$ CFU of *E. coli* K1 E11 Δ*aroA* or NaCl (*n* = 4) by intraperitoneal (IP) and subcutaneous routes (SC). **B** IgG response to three *E. coli* K1 strains initially selected for the study (E11 WT, S88 and RS218, responsible for neonatal meningitis) after each immunization by IP route (*n* = 4). **C** IgG response to *E. coli* K1 E11 WT and RS218 strains, after immunization with 3 doses by IP and SC route (*n* = 4). **D** Antibody titers after immunization with 3 doses (IP route on top, SC route in the bottom) (*n* = 4). **E** Detection of IgG1, Ig2a, IgG2b and IgG3 antibody isotypes after subcutaneous immunization in 3 doses (*n* = 4). **E** *p*-value values indicate significant differences in an ordinary one-way ANOVA. Error bars represent standard deviations (SD). Source data are provided as a Source Data file.

these K1 strains, with 24 h mortality rates of 0% (0 out of 4) in pups born to vaccinated mothers *versus* 100% (4 out of 4) in pups born to nonvaccinated mothers.

In 7 day-old pups born to mothers immunized with IP and challenged with three K1 strains ($10^6$ CFU of *E. coli* E11 WT, $10^6$ CFU of *E. coli* S88 and $10^6$ CFU of *E. coli* RS218 strains), we also observed strong protection with 24 h mortality rates of 0% (0 out of 4) in pups born to vaccinated mothers *versus* 100% (4 out of 4) pups born to

nonvaccinated mothers (Fig. 6D–F). The same complete protection was found for pups born to subcutaneously vaccinated mothers compared to pups born to unvaccinated mothers (Supplementary Fig. 4G–I). We also showed that the attenuated vaccine *E. coli* K1 E11 Δ*aroA* conferred the protection against two *E. coli* non-K1 strains (S242 and S510, $10^7$ CFU) infecting 7 day-old pups: survival rates of 100% were observed in pups born to immunized mothers. In contrast, the 24 h mortality rates in mouse pups born to non-immunized dams were

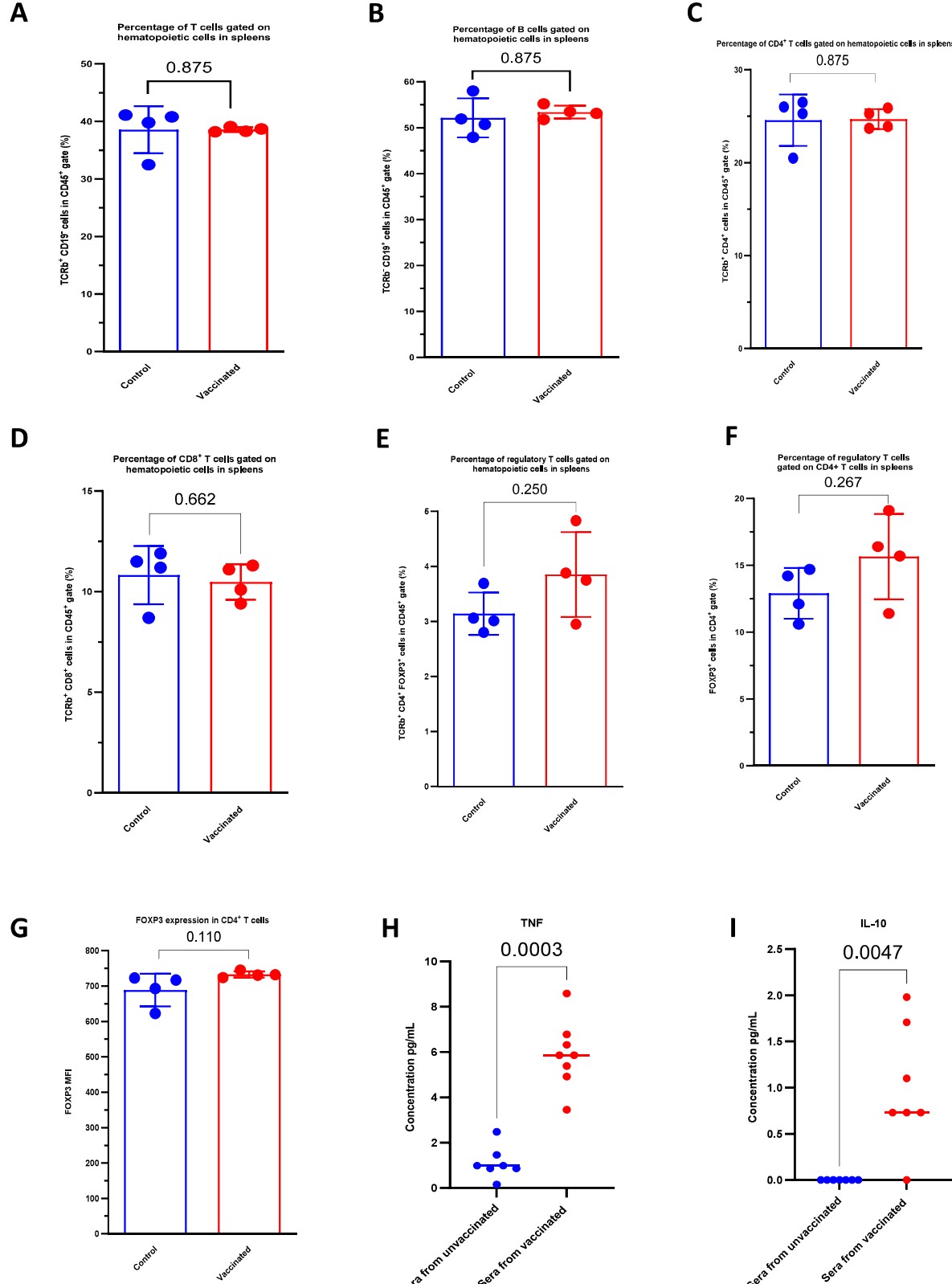

75% (3 out of 4) for *E. coli* S242 infections and 40% (2 out of 5) for S510 infections (Fig. 6G, H). While we had previously reported a high level of bacteria in the brains of pups after an intraperitoneal injection of the WT *E. coli* K1 E11[23], no bacteria were detected in the brains of the 7 day-old pups born to immunized mothers 24 h after the intraperitoneal injection (Fig. 6I).

Two weeks and 4 weeks old mice born to immunized mothers (IP route) were still completely protected against a lethal infection by *E. coli* K1 (Supplemental Fig. 3E, F).

Three-day-old pups mice born to mothers and immunized either with IP (Supplementary Fig. 3G, H) or SC (Supplementary Fig. 4J, K) were challenged by gavage with *E. coli* K1 strains. A 100% protection

**Fig. 3 | Evaluation of the cellular response after immunization with *E. coli* K1 E11 *ΔaroA*. A–E** Comparison of the expression of T cells (**A**), B cells (**B**), T cells CD4+ (**C**), T cells CD8+ (**D**) and regulatory T cells (**E**) among the hematopoietic splenocytes in the groups of vaccinated and non-vaccinated mice (*n* = 4). There were no significant intergroup differences in the various cell populations (*p* > 0.05). **F-G** Comparison of regulatory T cell expression (**F**) (*n* = 4) and FOXP3 expression by the T cell DC4+ population (**G**) (*n* = 4). The regulatory T cell count was higher in the vaccinated group of mice, but the difference was not significant (*p* > 0.05). The samples could be almost perfect superposed. **H–I** (*n* = 4): The concentration of TNF (**H**) and IL-10 (**I**) 7 days after the third dose was significantly higher in the sera of vaccinated mice than in nonvaccinated mice (*p* < 0.05). IL-2, IL4, IFN gamma and IL-17A were absent in the sera of the vaccinated and nonvaccinated groups. *P*-value values indicate significant and non-significant differences in a two-tailed Mann-Whitney non-parametric test (**A–I**). Error bars represent standard deviations (SD). Source data are provided as a Source Data file.

was found while a 50% mortality rate was observed in pups born from non-immunized mothers.

Taken as a whole, these in vivo results indicate that vertically transferred maternal antibodies can effectively protect newborns against severe infections caused by various *E. coli* K1and non-K1 strains.

## Discussion

Antibiotic-resistant *E. coli* strains have been found in premature infants and in women who have given birth prematurely[40,41]. The effectiveness of advanced antibiotic therapy for *E. coli* K1 meningitis is limited by the emergence of antibiotic resistance in some *E. coli* K1 strains[16,42,43]. A recent study in France found that 37% of the *E. coli* K1 strains causing meningitis produced an extended-spectrum beta lactamase and were resistant to third-generation cephalosporins[6]. Even when meningitis is caused by *E. coli* strains susceptible to antibiotics, the post-treatment mortality and sequalae rates are high[44,45]. In view of these ongoing issues, a vaccine would be a rapid, effective, preventive option. The effectiveness of a vaccine is driven by its safety and its immunogenicity. It is known that attenuated vaccines induce rapid, long-lasting, effective immunity[17,18,46,47]. Although live attenuated vaccines are not recommended during pregnancy, they can be considered in women planning to conceive so that an immune response can be transferred to the infant. The results of our study indicate that the *E. coli* K1 E11 *ΔaroA* strain is a good candidate for an attenuated vaccine with high levels of immunogenicity and strong preclinical in vitro and in vivo protection. We selected *E. coli* E11 K1 *ΔaroA* as the basis for our attenuated vaccine candidate because (i) we previously found that *aroA* is a virulence factor for *E. coli* K1, and (ii) this approach has been successful with other pathogens[26,27].

The specific immunity of newborns (including premature newborns) against pathogens relies mainly on maternal antibodies[48,49]. In contrast, maternal T cells are not involved because of maternal-fetal differences in tissue antigens (HLA in particular)[48]. We showed in vitro and in vivo the attenuation of the strain by deletion of the *aorA* gene. This attenuation is probably mediated for a significant part by the reduced expression of genes such as *FimH* encoding for the Type 1 fimbriae which is a well-recognized virulence factor for *E. coli* K1[31,32]. The role of the fimbriae probably play an important role in the results found in the adherence and invasion assays, however, we also have to take into account that type 1 fimbriae expression is controlled by a genetic switch that can easily been turned off or on[50]. Our present results showed that vaccination with *ΔaroA E. coli* E11 K1 induced a very strong humoral immune response and produced a very high level of bactericidal activity. We detected a higher level of IgG2 isotype after vaccination. Of note, IgG2 has a high capacity to bind to a novel gamma chain-dependent IgG activating Fc receptor, FcgammaRIV, and has high protective and pathogenic properties[51]. Our results also showed that the antibodies induced by immunization with this vaccine candidate fully protected female mice against various lethal doses of *E. coli* K1 and non-K1 strains. Other studies have shown that maternal antibodies are transferred to their offspring via the placenta and breast milk[49] and that these antibodies alone are enough to provide protection against perinatal infections[16].

In the present study, we showed that pups born to immunized mothers contained maternal polyclonal antibodies that were specific for *E. coli* K1. This antibody response was specific to the K1 and non-K1 *E. coli* strains responsible for meningitis. The fact that the female mice were vaccinated prior to mating demonstrated the efficacious, long-lasting immune protection induced by the *aroA* mutant *E. coli* K1 vaccine candidate.

Furthermore, our results indicated that pups born to vaccinated mothers were efficaciously protected, since both 3 day-old pups[52] and 7 day-old pups[45,53] were protected against lethal doses of various virulent *E. coli* strains causing meningitis. Furthermore, no bacteria were detected in the brains of pups born to vaccinated mothers. No detectable T cell responses were found in vaccinated mothers, but this did not influence the degree of protection observed in the pups.

Maternal antibodies can access fetal neural tissue through the developing blood-brain barrier and can thus prevent infection by vertically transmitted pathogens[32]. Indeed, the K1 and non-K1 *E. coli* strains responsible for meningitis are able to cross the blood-brain barrier and multiply in the cerebrospinal fluid[54].

Vaccination before or during pregnancy is useful in protecting newborn babies and pregnant women from serious infections[55]. The maternal antibodies induced by the vaccine can be transferred to the fetus and newborn via the placenta and breastfeeding, where they offer protection against specific micro-organisms. Maternal vaccination against the micro-organisms responsible for congenital and neonatal infections has therefore gain a major public health interest. Transplacental transfer of antibodies is a more natural, safer, more effective and less costly method of increasing neonatal antibody levels than administering immunoglobulin preparations to the infant[55,56]. However, the nature of the antigens and the type of vaccine administered during pregnancy also influence the effectiveness of antibody transfer to the fetus or newborn. Higher antibody titers have been observed in infants born to mothers immunized with conjugate vaccines compared with polysaccharide vaccines during pregnancy[56,57].

It has also been shown that asymptomatic carriage of *E. coli* K1 strains in the maternal gut microbiota is positively correlated with the risk of vertical transmission to the infant at birth[58,59]. Eliminating *E. coli* K1 strains from the maternal microbiota could also be a solution. Celine et al have shown that strains can be eliminated using specific phages[58]. Since bacteremia is a primordial stage in the pathophysiology of *E. coli* K1 meningitis[60,61], maternal vaccination induced by the transfer of specific IgG to newborns would be an excellent strategy for protecting newborns against neonatal *E. coli* K1 meningitis. Maternal vaccination could be an excellent way of better protecting offspring against neonatal *E. coli* K1 meningitis.

Our study shows that preconceptional vaccination with the *E. coli* K1 E11 *ΔaroA* vaccine induces strong immunity, driven by polyclonal bactericidal antibodies and protects not only mothers from infections during pregnancy, but also protects their offspring through vertical transmission of maternal immune effectors[55]. Maternal vaccination could be an excellent way of better protecting offspring against neonatal *E. coli* K1 infection and especially meningitis.

A limitation of our study is related to the transplacental transfer of IgG to the human fetus that occurs mainly during the last trimester of pregnancy and only reaches around 50% between 28- and 32 weeks' gestation[62]. Therefore, while late preterm and at term neonates could potentially benefit greatly from the strategy described in this work,

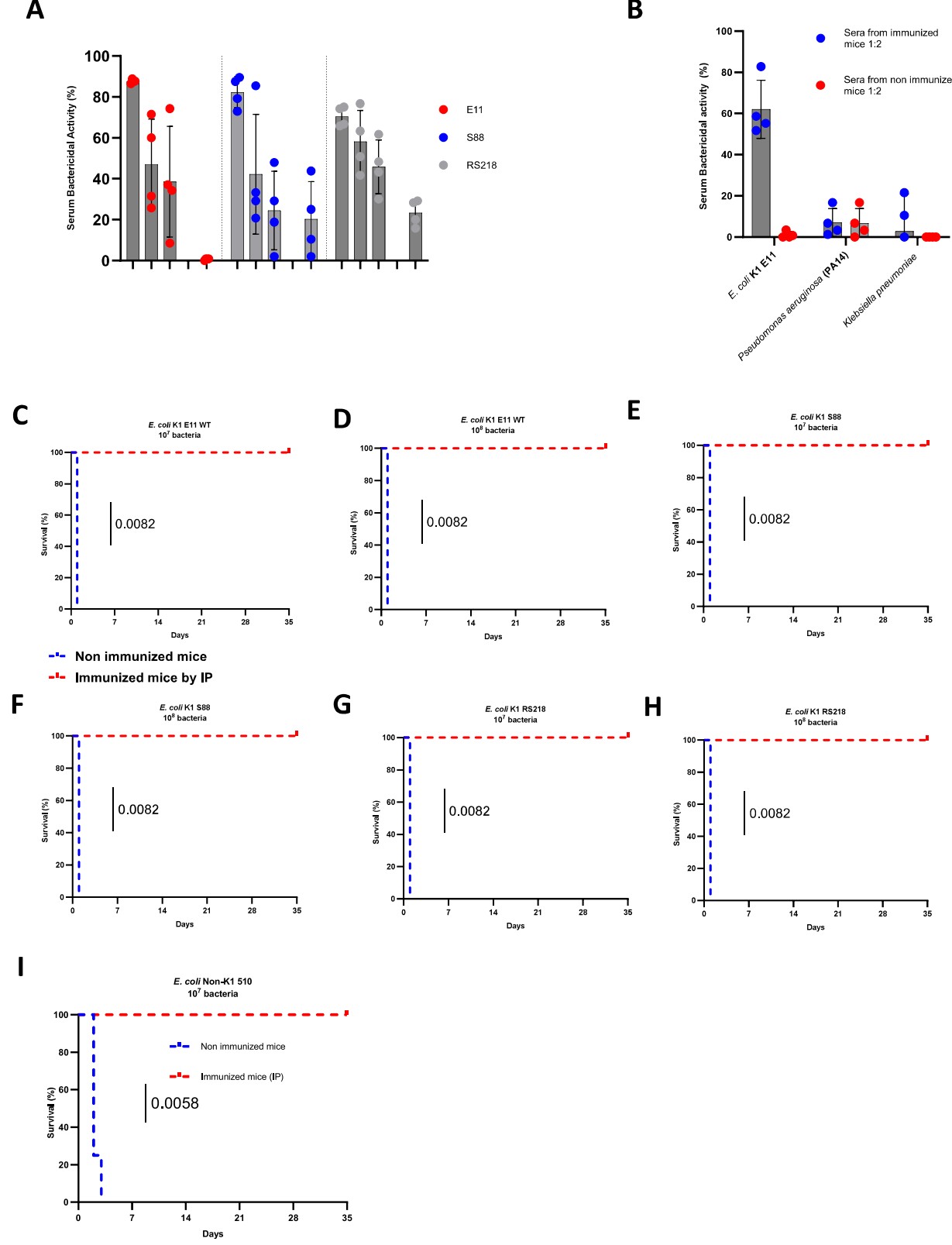

extremely preterm neonates born after <28 weeks gestation would probably benefit only to a limited extent from these protective maternal antibodies.

Another limitation of our study is the lack of evidence that prematurely born mice would be adequately be protected against severe *E. coli* infections. Further follow-up studies in larger preterm mammals,

beyond the scope of this current work, may be able to prove this concept in the future.

However, despite these limitations, our study provides an excellent preclinical proof of concept that maternal immunization with our attenuated live vaccine can protect neonates against severe *E. coli* infections.

**Fig. 4 | Evaluation of maternal protection by *E. coli* K1 E11 *ΔaroA* vaccine.**
**A** Percentage of bacterial activity obtained with various dilutions (1:2, 1:20 and 1:100) of sera from immunized and non-immunized mice (Normal Mice Serum, NMS), (*n* = 4) with the *E. coli* strains E11, S88 and RS218, using 3–4 week-old baby rabbit complement. **B** Percentage of activity obtained with sera from immunized mice or non-immunized mice with *Pseudomonas aeruginosa* PA14 and *Klebsiella pneumoniae*, using 3–4 week-old baby rabbit complement (*n* = 4). Bactericidal activity is considered potentially significant in vivo if in vitro killing is >30%. **C–I** On day 7 post-vaccination by the IP route, groups of vaccinated female BALB/c mice

(*n* = 4 per group) were challenged with lethal doses of 10⁷ CFU (**C**) and 10⁸ CFU (**D**) of *E. coli* K1 E11 WT, 10⁷ CFU (**E**) 10⁸ CFU (**F**) of *E. coli* K1 S88, 10⁷ CFU (**G**) 10⁸ CFU (**H**) of *E. coli* K1 RS218 and on day 7 post-vaccination by IP, groups of vaccinated female BALB/c mice (*n* = 4 per group) were challenged with lethal doses of 10⁷ CFU (**I**) of *E. coli* non-K1 S510. group of mice challenged with the S510 strain were immunized by IP. *P*-value indicate significant differences in a Mantel–Haenszel log-rank test. Error bars represent standard deviations (SD). Source data are provided as a Source Data file.

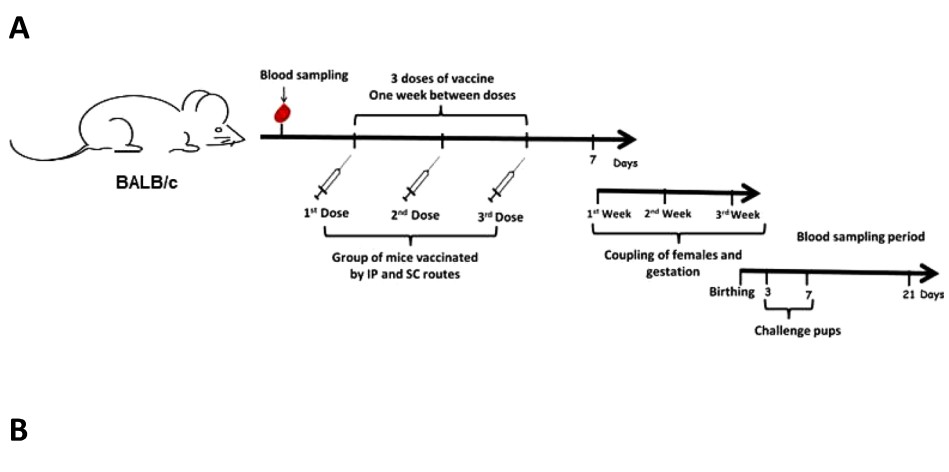

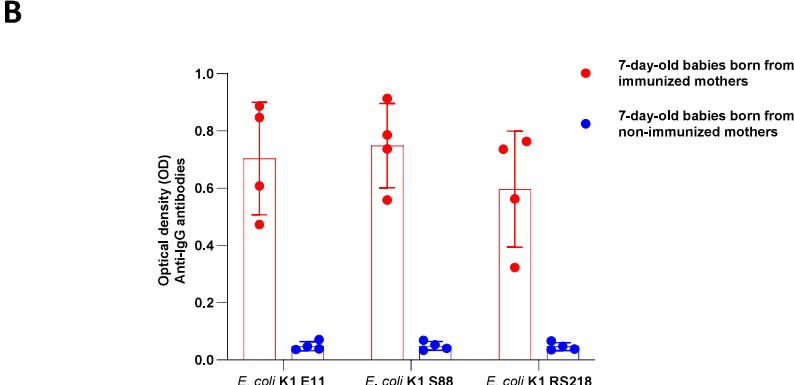

**Fig. 5 | Evaluation of the presence of specific maternal antibodies in mouse pups. A** Seven days after the last vaccination dose, immunized and non-immunized BALB/c female mice were mated with BALC/c males. Newborn mice born to immunized and non-immunized mothers were sacrificed at Day 7 (*n* = 4)

and their sera collected for (**B**) evaluation of the presence of maternal antibodies titers specific to *E. coli* K1 E11 WT, S88 and RS218 by whole-bacterial cell ELISA. Error bars represent standard deviations (SD). Source data are provided as a Source Data file.

Given the very high mortality rate and the neurological sequalae associated with neonatal *E. coli* K1 meningitis in preterm infants, our results constitute preclinical proof of concept for the development of a live attenuated vaccine against severe *E. coli* infections in women at risk of preterm delivery. The strategy we suggest, would be to vaccinate women at high risk of preterm delivery and more particularly those with a history of preterm delivery who envisage a new pregnancy. The benefits of such vaccination could be multiple. They mainly concern the newborns through passive protection against *E. coli* K1 infections, but potentially also by reducing preterm delivery related to maternal infection. Finally, our approach could also protect the mothers from *E. coli* sepsis during pregnancy.

Given that (i) women having already given birth prematurely have a very high risk of a subsequent preterm delivery (in turn associated with a greater risk of severe neonatal infections), (ii) the most common severe neonatal infection of this type is bacterial meningitis, and (iii) *E. coli* in general and *E. coli* K1 strains in particular constitute the most frequent cause of neonatal meningitis in this population, our results constitute strong preclinical proof-of-concept for the development of

a preconceptionally administered attenuated vaccine in this target population.

## Methods
### Ethical statement
The animal experiments were carried out at the Institut Imagine animal facility in Paris in compliance with current regulations (Directive 2010/63/EU of 22 September 2010 and Decree no. 2013-118 of 1 February 2013 on the protection of animals used for scientific purposes). The protocol was approved by an animal experimentation ethics committee approved by the French Ministry of Higher Education and Research (APAFIS number: 2020011519022360). Every effort has been made to minimize animal suffering. The animals come from an approved supplier and have SPF health status. SPF status considers the main pathogens in rodents and therefore makes it possible to establish a health status for "healthy" animals that are free from the main pathogens. On arrival, they are placed in a cage rack in an infectious A2 zone in cages containing nesting material and gnawing sticks. We also pay attention to social enrichment by limiting the isolation of our animals and we respect a 7 day acclamation period to allow the animals to

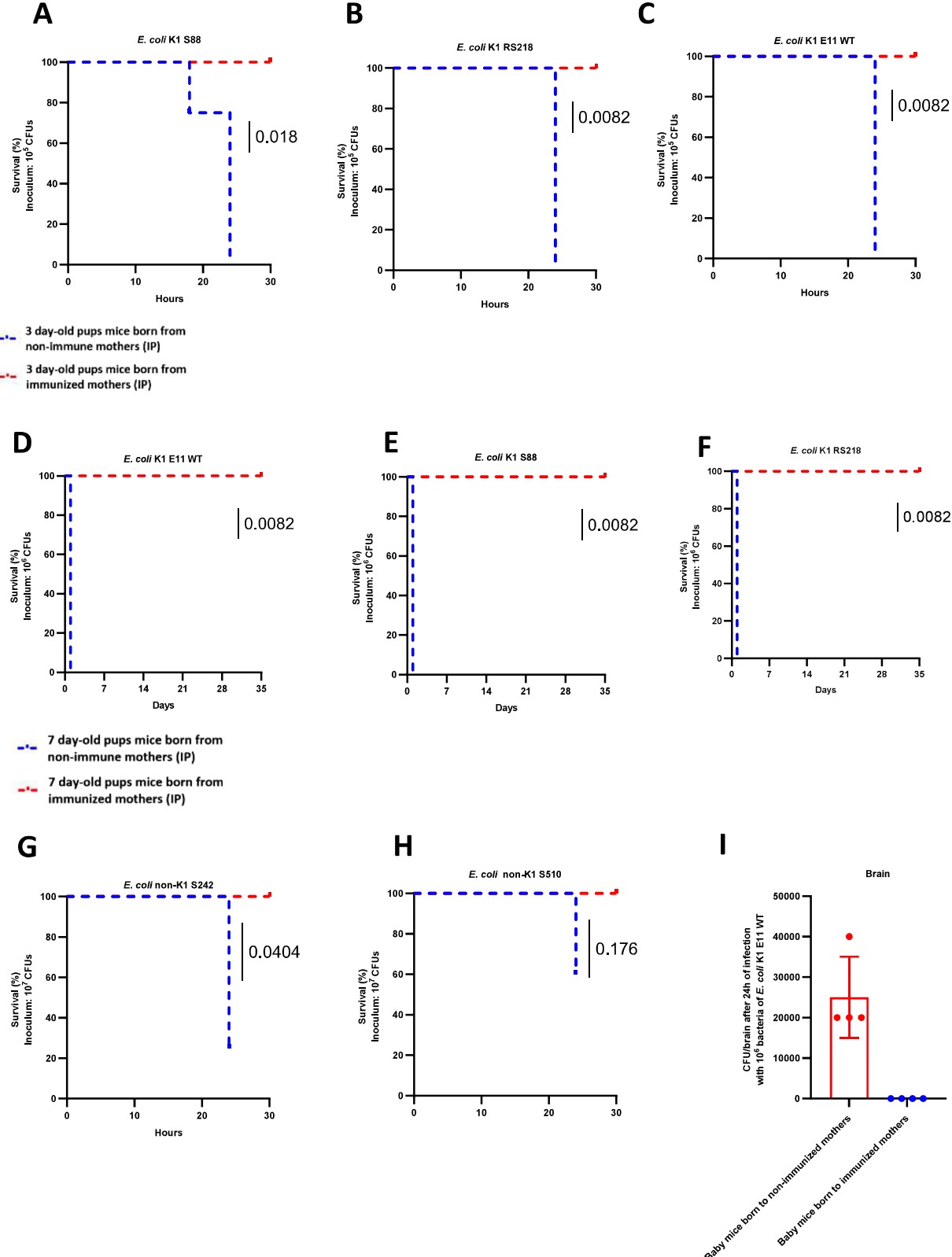

**Fig. 6 | Maternal immunization with *E. coli* K1 E11 *ΔaroA* protects against neonatal meningitis. A–C** Three days after delivery, groups of four pups born to immunized by IP and non-immunized female mice were injected intraperitoneally with $10^5$ CFU of *E. coli* K1 S88 (**A**), $10^5$ CFU of *E. coli* K1 RS218 (**B**), and $10^5$ CFU *E. coli* K1 E11 WT (**C**). Protection was assessed for 24 h. **D–F** Seven days after delivery, groups of four pups born to immunized by IP and non-immunized females were also given $10^6$ CFU *E. coli* K1 E11 WT (**D**), $10^6$ CFU *E. coli K1* S88 (**E**) and $10^7$ *E. coli* K1 RS218 (**F**).

Seven days after delivery, groups of four pups born to immunized and non-immunized by IP females were also given $10^7$ *E. coli* non-K1 S242 (**G**), and $10^7$ *E. coli* non-K1 S510 (**H**). **I** The CFU count in the brain 24 h after infection with $10^6$ CFU of *E. coli* K1 E11 WT bacteria in 7-day-old mouse pups. *P*-value values indicate significant and non-significant differences a Mantel–Haenszel log-rank test. Error bars represent standard deviations (SD). Source data are provided as a Source Data file.

recover from transport. The animals are also monitored every day, with appropriate limit points set up.

## Bacteria and cells

The *E. coli* K1 and *E. coli* non-K1 strains were provided by Professor Stéphane Bonacorci, head of the French national reference center for *E. coli* (Table 1). The *Hela* CRM-CCL-2™ cell line was provided by the ATCC and maintained at 37 °C with 5% CO$_2$ in Dulbecco's high glucose modified Eagle's medium (DMEM, Invitrogen, Carlsbad, CA, USA) with 10% fetal bovine serum (Dominique Dutscher, France).

## Construction of the aroA mutant

The 1284-nucleotide *aroA* gene is present in the *E. coli* K1 E11 strain. The methodology used to delete the *aroA* gene in this strain was described by Datsenko in 2000[33]. The technique is based on recombination between homologous sequences in the vicinity of the gene to be deleted and a previously generated DNA cassette. Recombination was facilitated by the Reda/Redb proteins encoded by the pRed plasmid. We used primers containing sequences (20 nucleotides) with homology to the kanamycin resistance cassette (1737 bp) contained in the genomic DNA of *E. coli* E11 *Δcsga* that we wished to amplify. These primers also contain 50-nucleotide sequences homologous to adjacent sequences to the *aroA* gene in *E. coli* E11 (Table 2). PCRs were used to generate a DNA product containing the kanamycin resistance cassette (1.7 kb), which was subsequently column-purified. The primer sequences are summarized in Table 2. The pRed plasmid was transferred into our *E. coli* strain by electroporation. Next, the strain was grown overnight on solid LB medium containing 100 µg/ml ticarcillin, in order to select bacteria containing the pRed plasmid. The resulting colonies were then re-isolated on LB agar plates containing 100 µg/ml ticarcillin and tested in a PCR assay with primers homologous to the ampicillin resistance gene contained in pRed. The proteins required for homologous recombination were induced by culturing *E. coli* plus pREd with L-arabinose. Next, the kanamycin resistance cassette with homology to sequences adjacent to the *aroA* gene was transferred into *E. coli* plus pRed by electroporation; the pRed plasmid is lost during this step following culture in liquid medium at 37 °C. Lastly, the bacteria were grown overnight on LB medium containing 20, 50 and 100 µg/ml kanamycin, in order to select mutants having received the kanamycin resistance cassette and lost the *aroA* gene through homologous recombination.

## Bacterial growth curve

The WT and *ΔaroA E. coli* strains were grown at 37 °C in LB liquid and DMEM medium. During this period, bacterial growth was measured in terms of the optical density (OD) every hour for 8 h. The WT and *ΔaroA E. coli* K1 E11 strains were grown under the same conditions.

## Growth in minimal medium M9

Wild type and *ΔaroA Escherichia coli* K1 E11 strains were grown for 48 h at 37 °C on M9 minimum medium (Sigma, France) with or without the

## Table 1 | Resource bacterial strains

| Strain | Type | Serogroups | K1 Ag | Phylogenetic group |
|---|---|---|---|---|
| S492 | Clinical isolate Neonatal meningitis | O45 | + | B21 |
| S495 | Clinical isolate Neonatal meningitis | O2 | + | B21 |
| S500 | Clinical isolate Neonatal meningitis | O18 | + | B21 |
| S509 | Clinical isolate Neonatal meningitis | O1 | + | B21 |
| S242 | Clinical isolate Neonatal meningitis | O25 | – | B2 |
| S395 | Clinical isolate Neonatal meningitis | O83 | + | B2 |
| S497 | Clinical isolate Neonatal meningitis | O6 | + | B2 |
| S503 | Clinical isolate Neonatal meningitis | O16 | + | B2 |
| S511 | Clinical isolate Neonatal meningitis | O75 | + | B2 |
| S482 | Clinical isolate Neonatal meningitis | O7 | + | F |
| S524 | Clinical isolate Neonatal meningitis | O1 | + | F |
| S510 | Clinical isolate Neonatal meningitis | O78 | – | C |
| S507 | Clinical isolate Neonatal meningitis | O12 | + | A |
| CFT073 | ATCC 700928 Pyelonephritis | O6 | – | B2 |
| S88 | Clinical isolate Neonatal meningitis | O45 | + | B21 |
| RS218 | ATCC 700973 Neonatal meningitis | O18 | + | B21 |
| E11 | Clinical isolate Neonatal meningitis | O18 | + | B21 |
| *Pseudomonas aeruninosa* PA14 | ATCC | – | – | – |
| *Klebsiella pneumoniae* | Clinical isolate | – | – | – |

## Table 2 | List of primers and probes used (Eurofins, France)

| Gene | Forward primer (5'-3') | Reverse primer (5'-3') |
|---|---|---|
| *aroA* (to generate the cassette by PCR) | GTTCATGGAATCCCTAAGGTTACAACCCATCGCTCGTGTCGATGGCACTA AATTAACCCTCACTAAAGGGCG | AATGTACCGCTAAAACTTTCCAGACTATTTCGAGCAATTGGCGCGGATA TATCCGCGCCAATTGCTCGAAATAGTCTGGAAAAGTTTTAGCGGTACATT TAATACGACTCACTATAGGGCTC |
| AroA E11 Upstream cassette kanamycin | TTTATAACGCCATGCCGCTG | CATGCTCCAGACTGCCTTG |
| Interne *aroA* E11 Upstream cassette kanamycin | CGAATGAAAGAACGCCCGAT | GCAAGAGGCGCAGTCATTAA |
| *aroA* Downstream kanamycin cassette | TTCATGATCTGTGTGTTGGTTTT | GCAATCGCCAGCAATAATTTAGT |
| *aroA* PCR verification of *aroA* deletion | GGTCCATTACACGCAGAAAG | GGAAACGGAGCCATCAACAT |
| *FimH* For RTqPCR | TGCAGAACGGATAAGCCGTGG | GCAGTCACCTGCCCTCCGGTA |
| *dnaE* | GATTGAGCGTTATGTCGGAGGC | GCCCCGCAGCCGTGAT |

addition of 40 µg/ml tryptophan, 40 µg/ml tyrosine, 40 µg/ml phenylalanine. A third condition with the addition of 2 times the amino acids was added. During this period, bacterial growth was measured by Optical Density.

### Bacterial adhesion assays with epithelial cells

The two strains of *E. coli* K1 E11 (WT and $\Delta aroA$) were cultured in LB liquid medium at 37 °C overnight. On the following day, the bacterial culture was diluted to obtain an $OD_{600}$ of 0.05 and then incubated at 37 °C until an $OD_{600}$ of between 0.6 and 0.7 was reached (corresponding to exponential growth). The strains were then pelleted by centrifugation at $5000 \times g$ for 10 min at 4 °C. The resulting supernatant was discarded, and the pellet was washed twice with PBS and resuspended in DMEM without antibiotics (Invitrogen, Carlsbad, CA, USA). In parallel, *Hela* epithelial cells were grown in 24-well plates at 37 °C in a 5% $CO_2$ humidified atmosphere for infection assays. The cells were then washed and incubated with the bacteria contained in the medium. The plates with infected *Hela* cells were placed at 37 °C in a humidified atmosphere at 5% $CO_2$ and washed five times with PBS, to eliminate non-adherent bacteria. The adherent bacteria present at the bottom of the wells were lysed with 500 µL of 0.2% Triton X-100 for 10 min on ice. The cell suspension was then serially diluted 10-fold with PBS and plated on LB agar plates to determine the adhesion frequency. For the invasion test, the content of each plate with the *Hela* epithelial cells was washed after 1 h of infection and then cultured in medium containing 100 µg/mL of gentamicin at 37 °C for 3 h. Next, we washed the plates five times with PBS, to eliminate non-adherent bacteria. The adherent cells present at the bottom of the wells were lysed with 500 µL of 0.2% Triton X-100 for 10 min on ice. The cell suspension was then serially diluted 10-fold with PBS and plated on LB agar plates to determine the invasion frequency.

### RNA-Seq analysis

*E. coli* K1 E11 strains WT and $\Delta aroA$ were grown in LB broth at 37 °C overnight. The following day, bacterial cultures of the 2 strains were grown from OD = 0.08 to exponential phage. At OD = 0.600, the bacteria were harvested and washed by centrifugation, then recovered in lysis buffer from the RNA extraction kit, Bioanalysis (Valencienner, Germany) reference 740955.50. Total RNA from both strains was extracted using this kit according to the manufacturer's instructions. RNA concentration was quantified using a spectrophotometer (Nanodrop Technologies). RNA-Seq analysis was carried out by the Genotyping/Sequencing Platform of the Brain Institute–ICM, France. mRNA library preparation was realized following manufacturer's recommendations (Stranded Total RNA Prep with Ribo-Zero Plus Kit from ILLUMINA). Final samples pooled library prep were sequenced on Nextseq 2000 ILLUMINA P2-200 cartridge (2x400Millions of 100 bases reads) corresponding to 2x14Millions of reads per sample after demultiplexing. RNA-Seq analysis was carried out by the Genotyping/Sequencing Platform of the Institut du Cerveau/Paris Brain Institute– ICM, France. Analysis of RNA sequencing data using CLC Genomics.

### RNA extraction and q-RTPCR

*E. coli* K1 E11 strains WT and $\Delta aroA$ were grown in LB broth at 37 °C overnight and in DMEM. The following day, bacterial cultures of the 2 strains were grown from OD = 0.08 to exponential phage. At OD = 0.600, the bacteria were harvested and washed by centrifugation, then recovered in lysis buffer from the RNA extraction kit, Bioanalysis (Valencienner, Germany) reference 740955.50. Total RNA from both strains was extracted using this kit according to the manufacturer's instructions. RNA concentration was quantified using a spectrophotometer (Nanodrop Technologies). For host gene expression, reverse transcription was performed using SuperScript™ III Platinum™ one-step RT-PCR (Life Technologies) and oligo(dT) primers. qPCR was performed using the Smart SYBRGreen fast Master kit

(Roche Diagnostics) and specific primers targeting the *FimH* gene (Table 2) in a CFX Touch detection system (Bio-Rad). Thermal cycling was performed at 55 °C for 10 min for RT, followed by 95 °C for 3 min, then 45 cycles at 95 °C for 15 s and 58 °C for 30 s. Measurement of quantified gene expression was expressed as cycle threshold (Ct) values normalized against the bacterial housekeeping gene *DnaE* (Table 2) and expressed as relative quantification (RQ) = 2 -ΔCt with ΔCt = Ct Target -Ct *DnaE*.

### Determination of the virulence of the WT and ΔaroA mutant strains

To assess the virulence of the WT and *ΔaroA E. coli* K1 E11 strains, we first determined the lethal dose by infecting 6 week-old female mice with different inoculums. For each strain, 20 mice were divided into five groups of four mice. The bacterial strains were grown in LB liquid medium up to an $OD_{600}$ of 0.6-0.8. Next, the bacteria were washed twice with sterile phosphate buffered saline (PBS) and resuspended in NaCl. Mice in each group received an intraperitoneal injection of $1 \times 10^9$, $1 \times 10^8$, $1 \times 10^7$, $1 \times 10^6$ or $1 \times 10^5$, CFU of bacteria in 200 μl of NaCl. The mortality rate was observed over a 1 week period.

### The antibody response of BALB/C female mice

To assess the immunogenicity of our *E. coli* K1 E11 *ΔaroA* strain against different clinical K1 and non-K1 *E. coli* strains, we performed a whole-bacterial-cell ELISA. First, we vaccinated a group of 6 week-old female mice with the mutant strain by the intraperitoneal (IP) and subcutaneous routes (SC) (Fig. 2A). In parallel, we administered a placebo (NaCl) to a control group. Blood samples were taken from the retro-orbital sinus 7 and 28 days after the last dose of vaccine or placebo. Antibody titers were assessed with the whole-bacterial-cell ELISA, according to standard methods[63]. To obtain heat-stable antigens from the various strains, bacteria were inactivated for 60 min in a water bath at 70 °C after an overnight culture in LB liquid at 37 °C. The culture media were then centrifuged at $20,000 \times g$ for 10 min in a microcentrifuge. The pellets were resuspended in PBS and diluted to an $OD_{600}$ of 0.1. Next, 100 μl of the different bacterial suspension were coated onto 96-well ELISA plates, which were incubated overnight at 37 °C. On the next day, 100 μl of bovine serum albumin (BSA) 1% were added to each well and incubated for 90 min at 37 °C. After washing with a mixture of PBS and Tween® 20 (Sigma-Aldrich, St. Louis, MO, USA), 100 μL of mice sera were added to each well and incubated for 2 h at 37 °C. Next, 100 μL of a 1:10000 dilution of anti-mouse secondary antibodies (rabbit anti-mouse IgG H&L, Abcam, Paris, France) coupled to horseradish peroxidase were added. A substrate was then used to reveal the reaction (TMB Liquid Substrate, TONBObioscience, San Diego, CA, USA).

Furthermore, we also inactivated the strain deleted from the *aorA* gene using a temperature of 70 °C for an hour, then immunized a group of mice with this strain and also evaluated their immunogenicity.

### SDS-PAGE gel electrophoresis and Western blots

The three *E. coli* K1 strains (E11 WT, S88 and RS218) and the two non-K1 strains (S242 and S510) were cultured at 37 °C until an $OD_{600}$ of 1.00 was reached. The culture media were centrifuged to collect the bacteria, which were then resuspended in 1 mL of RIPA lysis buffer (Sigma-Aldrich, St, Louis, MO, USA). Lysed samples were centrifuged, and 10 μL of the supernatant were electrophoresed in a 4–15% SDS-polyacrylamide gel (SDS-PAGE, BioRad, Marnes-la-Coquette, France). Proteins from gels not stained with Coomassie Blue reagent were transferred to nitrocellulose membranes using the Wet/Tank Blotting System and the Western Blotting Transfer System (BioRad, Marnes-la-Coquette, France), according to the manufacturer's instructions. Next, mouse sera were pooled according to their immunized and non-immunized status and placed in contact with the nitrocellulose

### Table 3 | List of antibodies used for flow cytometry experiments

| Antibodies | Provider | Reference | Dilution | Volume for 1 tube | Volume for 8 tubes |
|---|---|---|---|---|---|
| CD45 PerCP-Cy5.5 | BioLegend | 103132 | 1/100 | 1 uL | 8 uL |
| CD19 APCeF780 | eBioscience | 47-0193-80 | 1/100 | 1 uL | 8 uL |
| TCRb AF700 | BioLegend | 109224 | 1/50 | 2 uL | 16 uL |
| CD4 PB | BioLegend | 100534 | 1/100 | 1 uL | 8 uL |
| CD8 BV605 | BD Horizon | 563152 | 1/100 | 1 uL | 8 uL |
| FOXP3 APC | InVitrogen | 17-5773-82 | 1/20 | 5 uL | 40 uL |

membranes overnight at 4 °C. On the next day, anti-mouse secondary antibodies (rabbit anti-mouse IgG H&L (ab6709), Abcam, Paris, France) coupled to horseradish peroxidase were diluted (1:5000 in PBS) and used to detect primary antibodies. A chemiluminescent substrate (ECL Revel-Blot®, Ozyme, Saint-Cyr-l'Ecole, France) was used to reveal the presence of specific, polyclonal antibodies against bacteria proteins.

### Spleen cell phenotype analysis

Seven days after full immunization, BALB/c mice were sacrificed, and the spleens were collected for the flow cytometry analysis of lymphocytes. Red blood cells were removed from spleen cell suspensions by lysis with ammonium chloride buffer. The remaining cells were then incubated with LIVE/DEAD™ Fixable Aqua Dead Cell Stain Kit (Invitrogen L34957, Thermo Scientific Inc., Branchburg, NJ, USA) (Table 3) in PBS, according to the manufacturer's instructions. Next, cells were surface-stained with anti-CD45 PerCP-Cy5.5 (BioLegend, France), anti-CD19 APCeF780 (eBioscience 47-0193-80, France) (Table 3), anti-TCRb AF700 (BioLegend 109224, France), anti-CD8 BV605 (BD Horizon 563152), and anti-CD4 PB (100534) (Table 3) in cold PBS containing 0.5% BSA. The cells were then fixed and permeabilized with True-Nuclear™ Transcription Factor Buffer Set (BioLegend 424401, France), according to the manufacturer's instructions, and then stained with anti-FOXP3 allophycocyanin conjugate (Invitrogen 17-5773-82, Thermo Scientific Inc., Branchburg, NJ, USA) (Table 3) diluted in permeabilization buffer. Cells were then washed and resuspended in cold PBS containing 2% FCS. A Fortessa flow cytometer (BD Biosciences, San Jose, CA, USA) was used to analyze $10^5$ stained cells. Doublets were excluded from the analysis by using appropriate forward scatter/side scatter gates. Each immune subpopulation was defined as follows: B cells = CD45+ TCRb- CD19 + , T cells = CD45 + CD19- TCRb + , CD4 T cells = CD45 + CD19- TCRb+ CD8- CD4 + , CD8 T cells = CD45 + CDs19- TCRb+ CD4- CD8 + , and regulatory T cells = CD45 + CD19- TCRb + CD8- CD4 + FOXP3 + . Data were analyzed with Flowjo software. The cell count for a selected population was expressed as a percentage of the total hematopoietic cell count.

A gating strategy example is provided in Supplemental Fig. 6.

### Cytokine assays

Mouse cytokines (IL-10, IFN-γ, TNF-α, IL-4, IL-6, IL-2 and IL-17) in serum from immunized and non-immunized mice were assayed using the cytometric bead array (CBA) method (Becton Dickinson, Le Pont-de-Claix, France), according to the manufacturer's instructions. The CBA data were acquired with a BD Accuri™ cytometer (Becton Dickinson, Le Pont-de-Claix, France) and analyzed using HEC software. The results of the cytokine assays were expressed in pg/mL, as determined from a standard curve.

### The bactericidal activity assay

Serum bactericidal activity assays were performed to determine the activity of sera from immunized or non-immunized mice sera against

three strains of *E. coli* K1 (E11 WT, S88, and RS218), according to methods described previously[64,65]. Briefly, the strains were grown overnight at 37 °C in LB liquid media and then diluted to an $OD_{600}$ of 0.05. The dilutions were then incubated at 37 °C until an exponential growth was achieved ($OD_{600} \sim 0.7$). An appropriate volume of the culture medium was sampled to obtain $10^3$ CFU. The bacteria were then washed twice and resuspended in PBS, without additional $Mg^{2+}$ or $Ca^{2+}$. Next, 10 µl of the bacterial suspensions were added to 10 µl of heat-inactivated immunized sera, 10 µl of 3–4 week-old baby rabbit complement (Pel-Freez® Biologicals, AR, Rogers, USA), and 10 µl of PBS in a 96-well plate, and incubated at 37 °C. After a 1 h incubation, serial dilutions in PBS enabled us to determine the amounts of viable bacteria in each reaction mixture, with plating on solid LB medium overnight at 37 °C. In parallel, the initial bacterial concentration was determined by serial dilution. Bacterial killing was calculated by comparing the concentration of viable *E. coli* at the beginning of the assay with that at the end of the assay. As negative controls, assays were performed with heat-inactivated sera from non-immunized mice. Bactericidal activity >30% in two or more dilutions was considered as a positive assay.

### Evaluation of the protection conferred by the attenuated E. coli K1 E11 ΔaroA vaccine

To evaluate the degree of protection conferred by our live attenuated *E. coli* K1 E11 *ΔaroA* vaccine, several groups of 6 week-old female BALB/c mice were vaccinated (Fig. 2A). The vaccination schedule consisted of an intraperitoneal injection of a bacterial suspension ($10^5$ CFU of *E. coli* E11 *ΔaroA* in 200 µl of NaCl per adult female mouse) once a week for 3 weeks. One week after the third dose, the groups of mice received lethal doses ($10^7$ or $10^8$ CFU) of different *E. coli* K1 and non-K1 strains responsible for neonatal meningitis. Outcomes were monitored for 35 days.

### Evaluation of maternal vaccine antibody transfer and the protection of pups born to vaccinated mothers

To assess the protection of pup mice born to vaccinated mothers, groups of 6 week-old female BALB/c mice were immunized according to our vaccination schedule (Fig. 2A). In parallel, we administrated a same volume of a placebo (NaCl) to control mice. The different groups of female mice were matched with male mice. Three and 7 days after birth, two groups of four baby mice born to the vaccinated or non-vaccinated mothers as well as their mothers were sacrificed and their blood was collected to detect and quantify specific *E. coli* antibodies by whole-bacterial-cell ELISA and Western blots, respectively. Furthermore, groups of four pups (aged three and 7 days) from vaccinated or non-vaccinated mothers were given various doses of *E. coli* strains. The pups' survival at 24 h was evaluated.

### Data analysis

All data were analyzed using GraphPad Prism software (v9, GraphPad Software LLC, San Diego, CA, USA). Data are expressed as the mean ± standard deviation (SD). Groups were compared in a Mann-Whitney bilateral test, one-way ANOVA with correction for multiple comparisons or a Mantel–Haenszel log-rank test. The threshold for statistical significance was set to $p < 0.05$.

### Reporting summary

Further information on research design is available in the Nature Portfolio Reporting Summary linked to this article.

## Data availability

The authors declare that the data supporting the findings of this study are available within the article and its Supplementary Information files or are available from the authors upon request. Source data are provided with this paper. The RNAseq data have been deposited in NCBI. https://www.ncbi.nlm.nih.gov/geo/query/acc.cgi?acc=GSE252194. Source data are provided with this paper.

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

## Acknowledgements
We thank all the staff in the INEM group 1 and the staff in the Imagine Institute's animal house for their assistance. We also thank Samantha Schaeffer (INSERM UMRS 1138, "Functional genomics of solid tumors" group), Iris Delavaud stagaire student, Amaury Gensou (Janvier Labs—Faxility-Management, Paris, France) for their assistance and support. DS received an award from the ANR (AAPG2020-Seq-N-Vaq, France), and an award from the Charles H. Hood Foundation.

## Author contributions
Y.S., D.S. designed the experiments and interpreted the results. Y.S. lead all in vitro and in vivo experiments. C.S., H.F., M.A., E.L.W., Y.C.M., E.A., E.P., E.F., and F.B. carried out experiments and analyzed data. S.B. participated in the design of the study and provided bacterial strains. MC., VT., analyzed data and contributed to the drafting and revision of the manuscript. Y.S., D.S. wrote the first draft of the paper. All co-authors worked on iterations of the manuscript. D.S. provided the funding, developed the study concept and supervised the project.

## Competing interests
The authors declare no competing interests.
