## [Peer Review File · Nature Communications]

REVIEWER COMMENTS

Reviewer #1 (Remarks to the Author):

For obvious reasons, it is difficult to obtain data on the aetiology of neonatal bacterial meningitis and sepsis from patients, so most of our knowledge of these infections has been obtained from animal experimentation using rodents (often rats) or rabbits. The weight of evidence (some of it from human subjects) indicates that the new born infant obtains *E. coli* K1 from GI tract-colonised mothers; the bacteria then colonise the GI-tract of the infant and subsequently translocate from the child's gut lumen into the bloodstream via the local lymphatic system before crossing into the cerebrospinal fluid through the choroid plexus and invading the meninges. The bacteria are able to do this because the protective mucous layer in the small intestine has not matured at birth and gives the bacteria the opportunity to interact with the gut epithelium. The best infection models incorporate these aspects.

Prevention could be effected by eliminating K1 strains from the maternal GI microbiota and this would be good strategy to follow as a couple of studies in pregnant mothers have shown a correlation between neonatal infection and maternal K1 carriage. This might be readily done using K1-specific bacteriophage but surprisingly no one seems to have attempted this. Preventing K1 bacteria from traversing the neonatal gut might conceivably work but would probably require IgA antibody acquisition. Maternal vaccination has been mooted in the past but generally discounted as likely to be ineffective and difficult to achieve in large numbers of at-risk mothers; it would have been useful if the authors had discussed (or even mentioned) this literature.

The authors use the "unnatural" intraperitoneal route for the introduction of both the vaccine strain and the challenge bacteria to adult mice and conclude that their data provides "proof-of-concept for the development of a live attenuated vaccine against severe *E. coli* infections in women at risk of preterm delivery" but they offer no strategy for vaccine-derived prevention of infections in humans. Do they want to prevent infection in mothers or new born or both? How and by what route would the vaccine be administered? How frequently? Without some indication as to how this would be done in clinical practice their concluding statement is meaningless. For the last phase of the work they challenge 3- and 7-day-old pups with K1 strains but the route of administration of the challenge is not described in the methods section; I eventually found that they used the intraperitoneal route (probably very difficult to achieve in such small animals) from the legend to figure 6. This needs to be more prominently described in the methods and results sections as there are technical difficulties associated with this procedure in small animals. Figure 6 shows a significant difference in survival of pups from immunized mothers compared to non-immunized but they stop collecting data after 30 hours – what happened after this time point to convince that the difference was due to prevention and not just a delay in the lethal effect?

aroA encodes a gene involved in the biosynthesis of aromatic acids, so it does not directly impact on virulence. The authors found that the mutation reduces both adhesion and invasion so likely affects the expression of a number of virulence factors. It would be useful to have some data on the phenotype of the *aroA* mutant, in particular with regard to any changes in expression of the main K1 virulence

determinant, the polysialic acid capsule. The authors imply that protection may be afforded through the induction of antibodies that selectively activate the complement cascade to elicit a humoral bactericidal response, with little cellular immune response. They measure the IgG response in immunized adult mice (Figure 2) but IgM is a much more potent complement activator (which doesn't cross the placental barrier). Also, two of the four IgG classes are not complement activators and would therefore not contribute greatly to protection and may even block activation by other competent classes/subclasses. This needs to be discussed. Also, many K1 clinical isolates are complement-resistant and may therefore circumvent protection.

Reviewer #2 (Remarks to the Author):

This manuscript describes a series of experiments for the development and characterization of a preconceptionally administered maternal vaccine against neonatal *Escherichia coli* K1 infections in mice. The experiments were well conducted by an experienced team of researchers, using state-of-the-art methodologies, and support the conclusions of this study. Preterm births and *Escherichia coli* sepsis in preterm and term newborns are of major health concerns, with high mortality and organ injury among survivors. Developing a vaccine against *E. coli* K1 strains is therefore of high clinical priority. Using pre-conceptional maternal vaccination is an elegant and safer way of protecting the newborn through immunizations. The manuscript is therefore of high interest to the readership of the journal and to the wider scientific community.

The following changes or queries are suggested:

- Female mice were immunized via intraperitoneal injections of live-attenuated *E. coli*. Methods of microbial inoculation can impact the resulting host immune response. Intraperitoneal injections would not be suitable for human clinical use. Could the authors please comment on this potential limitation of their study, and have the authors conducted any experiments using alternative bacterial inoculation methods such as subcutaneous injections?
- Although live attenuated vaccines are generally more effective, as pointed out by the authors, inactivated microbes may be safer. Have the authors conducted any experiments on killed *E. coli* K1 E11 Δ aroA or other *E. coli* K1 strains, or have any further information on the immunogenicity of inactivated *E. coli* K1 strains that they could compare with their results?
- Do the authors have any information on the duration of immunity achieved through pre-conceptional immunization with the tested *E. coli* strain? What was the time difference between immunization and conception of the tested mice? Do the authors have any information on the relationship of timing of immunization and subsequent immune protection of the newborn?

- The authors used baby rabbit complement to assess the bactericidal activities against E coli strains in the serum of immunized adult mice. What was the rationale for using newborn rabbit complement rather than an adult complement preparation, which is expected to have higher complement activity?
- Title: Please consider indicating the use of mice in the title.
- Page 14 lines 308-309: Please check the following statement: 'For each strain, 40 mice were divided into five groups of four mice.'
- Page 22 line 520: 'Lane 5: the negative control.' Do the authors mean lane 6?
- Page 23 lines 526-527: 'In the gel profile, the left lane corresponds to the antigen lane and the left lane corresponds to the molecular weight standard.' Could the authors please clarify this statement?
- Figure 1 panels D to F: The different lines are overlapping and therefore difficult to distinguish, and the panels appear unclear. Please consider to revise these figure panels.
- Figure 2 panel B: The title of the panel reads 'E. coli non-K1', however the figure panel shows the results for E. coli K1 strains. Could the authors please clarify this aspect?
- Figure 2 panel D: The columns are very small and difficult to distinguish. Could the authors please enlarge the panel elements?
- Figure 3 legend: 'A trend of increased regulatory T was observed in the vaccinated group of mice but was significantly different between the two groups ($P > 0.05$). Panel G does not indicate any significant group difference. Could the authors please clarify or correct the discrepancy?
- Figure 4: Could the authors please use the same font size for the panel titles.
- Figure 5: 'Day' appears to be 'postnatal day'; could the authors please clarify this information?
- Figure 6 panel A: Please add the asterisks for significance.
- Figure 6 panel H: The y-axis displays CFU/ml in the brain of neonatal mice. Since these are CFU in tissue, how did the authors arrive at CFU per ml?
- Supplementary figure 2 panel E: These seem to be sera of baby mice from immunized dams.
- Supplementary figure 2: Could the authors please clarify what is meant by detection of anti-mouse IgG antibodies?
- Page 12 line 273: Do the authors mean xylazine?
- Page 20 line 471, also page 34 figure 3 legend: Please change 'DC4+ T cell' to 'CD4+ T cell'.

Reviewer #3 (Remarks to the Author):

The context provided for the research is development of a vaccine to prevent or limit impact of neonatal *E. coli* infections, potentially targeting pregnant individuals who have already had one pre-term deliver. The work provides a clear demonstration of how immunisation of mice with an *aroA* mutant of an *E. coli* K1 strain can provide protection of those mice from *E. coli* challenge with either the parental strain or other *E. coli*. The work demonstrates protection of pups by passive transfer from female mice vaccinated before mating. The work supports the findings with analysis of humoral and T-cell responses.

The concept of vaccination to protect against *E. coli* and other bacterial infections with killed or attenuated strains is very well established as is protection by passive transfer from vaccinated females. This is particularly the case for infections in pigs and cattle with some demonstration in mouse models. Papers dating back to 1970s include: PMID: 7679376 PMID: 2410365 PMID: 31985 PMID: 37031607 PMID: 25475689 PMID: 24797699 PMID: 22612965 PMID: 2497598 PMID: 3057757 PMID: 364066 PMID: 6347759. *AroA* mutation is also an established way to attenuate strains for live vaccines (>300 publications) as it results in slower growth under *in vivo* conditions due to incapacity to make aromatic amino acids although there has been one report of increased virulence (PMID: 27601574). The general route of the protection is well understood to be transfer of antibody in colostrum and not cell mediated. So for presented manuscript all the steps presented are known and understood but it is still important to show the concept can be realised of *E. coli* K1 associated with serious neonatal infections, so in relation to this specific context there is some novelty (as asked of the reviewer to assess) although good previous literature on K1 strain vaccine development in relation to meningitis (K S Kim and Tim Korhonen).

On the whole the data is well presented with appropriate statistical analyses and interpretation. My main issue is the use of the 24 hr cut off for the pup infections coming from vaccinated and unvaccinated animals (Figure 6). Clearly there is an important difference at this time point but later time points are not shown. Is this because that data was not collected or that pups do begin to succumb to infection after 24hr? Much longer running animal expts following infection are used earlier in the study (Figure 4) so it is not clear why this cut-off is used here? With this type of protection, it is directly related to circulating antibody from the transfer and a direct inoculation of 10⁵ bacteria is a serious challenge for the pups, so if there is additional information about the protection for the different strains at longer assessment points then it is still important information to include, otherwise the 24hr cut-off needs to be justified.

The description of *aroA* as a virulence determinant is also questionable. I appreciate the authors show the reduced binding and invasion of the mutant compared to wild type with equivalent growth of the two in LB, however it is known that the issue of an *aroA* mutant comes with conditions under more restricted growth conditions when the bacteria has to synthesize key amino acids from scratch which may be in very short supply *in vivo*. As such the best growth conditions to support a role in virulence vs growth would be the tissue culture medium/media used in the binding and invasion assays rather than LB?

REVIEWER COMMENTS

Reviewer #1 (Remarks to the Author):

*“For obvious reasons, it is difficult to obtain data on the aetiology of neonatal bacterial meningitis and sepsis from patients, so most of our knowledge of these infections has been obtained from animal experimentation using rodents (often rats) or rabbits. The weight of evidence (some of it from human subjects) indicates that the new born infant obtains *E. coli* K1 from GI tract-colonised mothers; the bacteria then colonise the GI-tract of the infant and subsequently translocate from the child’s gut lumen into the bloodstream via the local lymphatic system before crossing into the cerebrospinal fluid through the choroid plexus and invading the meninges. The bacteria are able to do this because the protective mucous layer in the small intestine has not matured at birth and gives the bacteria the opportunity to interact with the gut epithelium. The best infection models incorporate these aspects.”*

We agree, and infection by gavage with *E. coli* K1 of the new born mice is the main route of infection we used in our previous work that identified AroA as a virulence factor (EBioMedicine. 2023 Feb;88:104439). In this previous work we used both the IP route and the gavage route (that starts with the GI tract colonization described by the reviewer). As the IP route is more severe (Figure S5 from this previous work), we selected this route of infection for the proof of concept efficacy of our approach with the Δ *aroA* vaccine, inferring that a protection against the most severe IP model would mean a de facto infection against the gavage model too. In addition, as previously published, at D7 we can not infect the pups by gavage (the animals are protected in grand part because of the better production of mucus preventing translocation from the GI tract). However, as the reviewer is right and the gavage model is probably closer to the human infection, we have now added protection against different *E. coli* K1 strains using the gavage model at D2-D3 (Three-day-old pups mice born to mothers and immunized either with IP (**Supplementary figure 3G-H**) or SC (**Supplementary figure 4J-K**)).

“Prevention could be effected by eliminating K1 strains from the maternal GI microbiota and this would be good strategy to follow as a couple of studies in pregnant mothers have shown a correlation between neonatal infection and maternal K1 carriage.

This might be readily done using K1-specific bacteriophage but surprisingly no one seems to have attempted this. Preventing K1 bacteria from traversing the neonatal gut might conceivably work but would probably require IgA antibody acquisition. Maternal vaccination has been mooted in the past but generally discounted as likely to be ineffective and difficult to achieve in large numbers of at-risk mothers; it would have been useful if the authors had discussed (or even mentioned) this literature.”

This is now discussed in the manuscript including the strategy using phages to eliminate *E. coli*. (References: 54 and 55).

“The authors use the “unnatural” intraperitoneal route for the introduction of both the vaccine strain and the challenge bacteria to adult mice and conclude that their data provides “proof-of-concept for the development of a live attenuated vaccine against severe E. coli infections in women at risk of preterm delivery” but they offer no strategy for vaccine-derived prevention of infections in humans.

Do they want to prevent infection in mothers or new born or both? How and by what route would the vaccine be administered? How frequently? Without some indication as to how this would be done in clinical practice their concluding statement is meaningless. “

We agree with the reviewer are we are now presenting a new set of data using a more translatable route of immunization: the subcutaneous injection. Both mother and new-born mice were still protected with this new vaccination regimen. Results for protection against three different strains of *E. coli* K1 are now presented after SC immunization.

This new regimen is a three-dose subcutaneous vaccination with a one-week interval between doses. We now show:

- (1) the presence of antibodies specific to *E. coli* K1 in immunized female mice (**Figure 2C, D et E**),
- (2) protection of the immunized females against infections with *E. coli* K1 strains (**Supplementary figure 4A, 4B and 4C**)
- (3) protection of pups born from mother immunized SC against infections with *E. coli* K1 strains using two route of infection: IP and gavage (**Supplementary figure 4D, E, F, G, H, I, J, K**).

“For the last phase of the work they challenge 3- and 7-day-old pups with K1 strains but the route of administration of the challenge is not described in the methods section; I eventually found that they used the intraperitoneal route (probably very difficult to achieve in such small animals) from the legend to figure 6. This needs to be more prominently described in the methods and results sections as there are technical difficulties associated with this procedure in small animals. Figure 6 shows a significant difference in survival of pups from immunized mothers compared to non-immunized but they stop collecting data after 30 hours – what happened after this time point to convince that the difference was due to prevention and not just a delay in the lethal effect?”

The route of infection of the baby mice is now clearly shown in the manuscript, both in the materials and methods section and in the results.

When baby mice are infected at D2-D3, the mother is eating the infected pups, that why we separate the mother from the mice after infection. Then we have to sac the pups at 24-30h to avoid starvation.

However, this issue is not here when the infection is done at D7. Therefore, we can now present the data at much longer times: (35 days post-infection) (**Figure 6D, E, F** for IP) and (**Supplementary figure 4G, H, I** for SC).

New and longer time points have also been added:

-We show new protection data for immunized adults: protection is maintained at 60 days: **Supplemental Figure 3B-D)**

-Pups born from immunized mother are still protected against infection at 14days and 28days: **Supplemental Figure 3E-F**

We have also added data from the infection of baby D3 mice by gavage (**Supplementary figure 3G and H**) and (**Supplementary figure 4J and 4K**)

“aroA encodes a gene involved in the biosynthesis of aromatic acids, so it does not directly impact on virulence. The authors found that the mutation reduces both adhesion and invasion so likely affects the expression of a number of virulence factors. It would be useful to have some data on the phenotype of the aroA mutant, in particular with regard to any changes in expression of the main K1 virulence determinant, the polysialic acid capsule.”

As suggested by the reviewers 1 and 3 we further studied the role of *aroA* as a virulence determinant.

-We added the growth conditions used for adhesion and invasion assays (DMEM) (**Figure 1A**) and even studied the impact of the deletion of *aroA* at the full genome scale by performing RNAseq experiments comparing *E. coli* E11WT and *E. coli* Δ *aroA* in LB and DMEM. The main results from these experiments are now presented **Figure 1A, 1D** and **Table supplemental 1**.

A striking result was the impact of the genes for the type 1 fimbriae, a major virulence factor for *E. coli* K1 (**supplemental Figure 5A, B** and **Table supplemental 2**). This surprising result provide a better understanding on the decrease in vitro and in vivo virulence of the Δ *aroA* strain. This result was confirmed by RTqPCR (**Figure 1E**)

“The authors imply that protection may be afforded through the induction of antibodies that selectively activate the complement cascade to elicit a humoral bactericidal response, with little cellular immune response. They measure the IgG response in immunized adult mice (Figure 2) but IgM is a much more potent complement activator (which doesn't cross the placental barrier). Also, two of the four IgG classes are not complement activators and would therefore not contribute greatly to protection and may even block activation by other competent classes/subclasses. This needs to be discussed.”

As suggested by the reviewer we have extended this part of our study, and have added data evaluating the levels of three IgG isotypes (Ig1, Ig2 and Ig3). We detected a higher level of IgG2 isotype after vaccination. It was previously shown that IgG2 has a high capacity to bind

to a novel gamma chain-dependent IgG activating Fc receptor, FcγR4, and has high protective and pathogenic properties. (Reference 46)

“Also, many K1 clinical isolates are complement-resistant and may therefore circumvent protection.”

We added data on the bactericidal activity of the serum with two other strains of *E. coli* K1 (S524 and S395 (**Supplementary figure 3A**) but mentioned in the text that other strains could be resistant in this assay (with a ref associated).

However, in vivo, the killing of microorganisms by maternal antibodies is not mediated only by complement. We and other have reported reviews summarizing the literature on these mechanisms and this review has also been added to the text (Reference 29).

Reviewer #2 (Remarks to the Author):

“This manuscript describes a series of experiments for the development and characterization of a preconceptionally administered maternal vaccine against neonatal Escherichia coli K1 infections in mice. The experiments were well conducted by an experienced team of researchers, using state-of-the-art methodologies, and support the conclusions of this study.”

We thank the reviewer for this positive comment

“Preterm births and Escherichia coli sepsis in preterm and term newborns are of major health concerns, with high mortality and organ injury among survivors. Developing a vaccine against E. coli K1 strains is therefore of high clinical priority. Using pre-conceptual maternal vaccination is an elegant and safer way of protecting the newborn through immunizations. The manuscript is therefore of high interest to the readership of the journal and to the wider scientific community.

The following changes or queries are suggested:

• Female mice were immunized via intraperitoneal injections of live-attenuated E. coli. Methods of microbial inoculation can impact the resulting host immune response. Intraperitoneal injections would not be suitable for human clinical use. Could the authors please comment on this potential limitation of their study, and have the authors conducted any experiments using alternative bacterial inoculation methods such as subcutaneous injections?”

We added data from the three-dose subcutaneous vaccination with a one-week interval between doses. We showed: (1) the presence of antibodies specific to *E. coli* K1 in immunized female mice (**Figure 2C, D et E**), (2) these antibodies were able to protect these

females against infections with *E. coli* K1 strains (**Supplementary figure 4A, 4B and 4C**) (3) we then showed that maternal vaccine antibodies passively transmitted to baby mice were able to protect them against infections with *E. coli* K1 strains (**Supplementary figure 4D, E, F, G, H, I, J, K**).

• “Although live attenuated vaccines are generally more effective, as pointed out by the authors, inactivated microbes may be safer. Have the authors conducted any experiments on killed *E. coli* K1 E11 Δ aroA or other *E. coli* K1 strains, or have any further information on the immunogenicity of inactivated *E. coli* K1 strains that they could compare with their results?”

We inactivated the *E. coli* K1 strain and compared the immunogenicity to the attenuated strain, we observed a significant difference in the level of specific antibodies with the live attenuated strain significantly more immunogenic than the inactivated strain (**Supplementary figure 2J**). In fact, the peak of antibodies detected after immunization with the inactivated strain is as low as the level of antibodies detected 30 days after immunization with the live attenuated vaccine (**Figure 2D**)

• “Do the authors have any information on the duration of immunity achieved through pre-conceptional immunization with the tested *E. coli* strain? What was the time difference between immunization and conception of the tested mice? Do the authors have any information on the relationship of timing of immunization and subsequent immune protection of the newborn?”

We have added a new, well-detailed vaccination schedule (**Figure 5A**).

(1) Female mice are immunized in three doses with the Δ aroA vaccine with an interval of one week between doses.

(2) 7 days after the last doses, the females are paired with the males.

(3) 21 days from the first day of mating, the baby mice are born.

(4) The baby mice of 3 and 7 days are challenged.

We also present new data on the duration of the immunity with a maintained protection against lethal infection of pups born from immunized mice at D14 and D28 days (**Supplementary figure 3E et F**).

• “The authors used baby rabbit complement to assess the bactericidal activities against *E. coli* strains in the serum of immunized adult mice. What was the rationale for using newborn rabbit complement rather than an adult complement preparation, which is expected to have higher complement activity?” a

We use baby rabbit complement because it has long been the standard complement option recommended by the World Health Organization (WHO) for functional antibody testing to assess vaccine effectiveness.

(<https://doi.org.proxy.insermbiblio.inist.fr/10.1016/j.vaccine.2007.04.043>),

<https://www.who.int/publications/m/item/meningococcal-polysaccharide-vaccine-annex-6-trs-no-658>

- Title: Please consider indicating the use of mice in the title.

We understand the reviewer comment. As pre-clinical is already in the title we may want to keep it simple but if necessary, the title could be changed.

- Page 14 lines 308-309: Please check the following statement: 'For each strain, 40 mice were divided into five groups of four mice.'

Done

- Page 22 line 520: 'Lane 5: the negative control.' Do the authors mean lane 6?

Done

- Page 23 lines 526-527: 'In the gel profile, the left lane corresponds to the antigen lane and the left lane corresponds to the molecular weight standard.' Could the authors please clarify this statement?

Lane M corresponds to the molecular weight, and lanes 1, 2, 3, 4 and 5 correspond to the clone of bacteria tested (samples).

- Figure 1 panels D to F: The different lines are overlapping and therefore difficult to distinguish, and the panels appear unclear. Please consider to revise these figure panels.

We enlarged the points.

E11: Lethal dose 10^7

S88: Lethal dose 10^7

RS218: Lethal dose 10^6

S242: Lethal dose 10^7

S510: Lethal dose 10^7

Δ aroA: 10^9 only 25% mortality

- Figure 2 panel B: The title of the panel reads 'E. coli non-K1', however the figure panel shows the results for E. coli K1 strains. Could the authors please clarify this aspect?

Done: Modified

- Figure 2 panel D: The columns are very small and difficult to distinguish. Could the authors please enlarge the panel elements?

Done

- Figure 3 legend: ‘A trend of increased regulatory T was observed in the vaccinated group of mice but was significantly different between the two groups ($P>0.05$). Panel G does not indicate any significant group difference. Could the authors please clarify or correct the discrepancy?’

We have added: but with no significant difference.

- Figure 4: Could the authors please use the same font size for the panel titles.

Done

- Figure 5: ‘Day’ appears to be ‘postnatal day’; could the authors please clarify this information?’

Done

- Figure 6 panel A: Please add the asterisks for significance.

Done

- Figure 6 panel H: The y-axis displays CFU/ml in the brain of neonatal mice. Since these are CFU in tissue, how did the authors arrive at CFU per ml?

We put CFU/brain

- Supplementary figure 2 panel E: These seem to be sera of baby mice from immunized dams.

Panel E is the serum from adult females and panels G and H are the sera from baby mice from immunized mothers.

- Supplementary figure 2: Could the authors please clarify what is meant by detection of anti-mouse IgG antibodies?’

This is modified: anti-*E.coli* K1 IgG antibodies

- Page 12 line 273: Do the authors mean xylazine?’

Yes, xylazine.

- Page 20 line 471, also page 34 figure 3 legend: Please change ‘DC4+ T cell’ to ‘CD4+ T cell’.

Done

Reviewer #3 (Remarks to the Author):

“The context provided for the research is development of a vaccine to prevent or limit impact of neonatal E. coli infections, potentially targeting pregnant individuals who have already had one pre-term deliver. The work provides a clear demonstration of how immunisation of mice with an aroA mutant of an E. coli K1 strain can provide protection of those mice from E. coli challenge with either the parental strain or other E. coli. The work demonstrates protection of pups by passive transfer from female mice vaccinated before mating. The work supports the findings with analysis of humoral and T-cell responses.”

We thank the reviewer for the positive comment

“The concept of vaccination to protect against E. coli and other bacterial infections with killed or attenuated strains is very well established as is protection by passive transfer from vaccinated females. This is particularly the case for infections in pigs and cattle with some demonstration in mouse models. Papers dating back to 1970s include: PMID: 7679376 PMID: 2410365 PMID: 31985 PMID: 37031607 PMID: 25475689 PMID: 24797699 PMID: 22612965 PMID: 2497598 PMID: 3057757 PMID: 364066 PMID: 6347759. AroA mutation is also an established way to attenuate strains for live vaccines (>300 publications) as it results in slower growth under in vivo conditions due to incapacity to make aromatic amino acids although there has been one report of increased virulence (PMID: 27601574). The general route of the protection is well understood to be transfer of antibody in colostrum and not cell mediated. So for presented manuscript all the steps presented are known and understood but it is still important to show the concept can be realised of E. coli K1 associated with serious neonatal infections, so in relation to this specific context there is some novelty (as asked of the reviewer to assess) although good previous literature on K1 strain vaccine development in relation to meningitis (K S Kim and Tim Korhonen).“

We have now added these references to the manuscript.

“On the whole the data is well presented with appropriate statistical analyses and interpretation. My main issue is the use of the 24 hr cut off for the pup infections coming from vaccinated and unvaccinated animals (Figure 6).”

When baby mice are infected at D2-D3, the mother is eating the infected pups, that why we separate the mother from the mice after infection. Then we have to sac the pups at 24-30h to avoid starvation.

However, this issue is not here when the infection is done at D7. Therefore, we can now present the data at much longer times: (35 days post-infection): (**Figure 6D, E, F for IP**) and (**Supplementary figure 4G, H, I for SC**)

“Clearly there is an important difference at this time point but later time points are not shown. Is this because that data was not collected or that pups do begin to succumb to infection after 24hr? Much longer running animal expts following infection are used earlier in the study (Figure 4) so it is not clear why this cut-off is used here? With this type of protection, it is directly related to circulating antibody from the transfer and a direct inoculation of 10(5) bacteria is a serious challenge for the pups, so if there is additional information about the protection for the different strains at longer assessment points then it is still important information to include, otherwise the 24hr cut-off needs to be justified”.

Mother eat the pups that are sick. When baby mice are infected at D2-D3, the mother is eating the infected pups, that why we separate the mother from the mice after infection. Then we have to sac the pups at 24-30h to avoid starvation.

However, this issue is not here when the infection is done at D7. Therefore, we can now present the data at much longer times: 35 days post-infection (**Figure 6D, E, F for IP**) and (**Supplementary figure 4 G, H, I for SC**).

*“The description of *aroA* as a virulence determinant is also questionable. I appreciate the authors show the reduced binding and invasion of the mutant compared to wild type with equivalent growth of the two in LB, however it is known that the issue of an *aroA* mutant comes with conditions under more restricted growth conditions when the bacteria has to synthesize key amino acids from scratch which may be in very short supply in vivo. As such the best growth conditions to support a role in virulence vs growth would be the tissue culture medium/media used in the binding and invasion assays rather than LB?”*

As suggested by the reviewers 1 and 3 we further studied the role of *aroA* as a virulence determinant.

-We added the growth conditions used for adhesion and invasion assays (DMEM) (**Figure 1A**) and even studied the impact of the deletion of *aroA* at the full genome scale by performing RNAseq experiments comparing *E. coli* E11WT and *E. coli* Δ *aroA* in LB and DMEM. The main results from these experiments are now presented **Figure 1A, 1D and Table supplemental 1**.

A striking result was the impact of the genes for the type 1 fimbriae, a major virulence factor for *E. coli* K1. This surprising result provide a better understanding on the decrease in vitro and in vivo. This result was confirmed by RTqPCR (**Figure 1, supplemental Figure 5A, B and Table supplemental 2, Figure 1E**).

REVIEWER COMMENTS

Reviewer #2 (Remarks to the Author):

As previously stated, this is a highly relevant topic of scientific and clinical relevance, and experiments were well conducted and are supporting the conclusions. The authors have added several additional informative experiments in support of their studies as well as additional references and discussion points, and as such their revised manuscript version is significantly improved and adequately addresses most of the reviewers' critiques. Nevertheless, I do have a few remarks.

1) Introduction -page 3 lines 49-50: "With a death rate of 35% for preterm newborns and 18% for preterm children up until the age of 5 years, prematurity is a true public health problem worldwide 2,3."

Per reference 3 (Walani), "... preterm birth is the leading cause of death among children, accounting for 18% of all deaths among children aged under 5 years and as much as 35% of all deaths among newborns (aged <28 days)." This means that 35% of all deaths among newborns are due to prematurity, rather than that 35% of all preterm newborns do not survive. Please correct this statement.

2) I would also like to remark the following: The authors state that "women with a history of preterm delivery have a high risk of recurrence and therefore constitute a target population for the development of a vaccine against neonatal E. coli infections." Undoubtedly preterm neonates, especially very preterm neonates, are at high risk for severe infections such as E. coli sepsis and meningitis. The authors show that immunization of female mice with the attenuated E. coli K1 E11 Δ aroA vaccine leads to transfer of protective IgG antibodies to the pups. Transplacental IgG transfer to the human fetus takes primarily place during the last trimester of pregnancy, reaching only about 50% by 28 to 32 weeks gestation (e.g., PMID: 22235228). While late preterm and term newborns could potentially greatly benefit from this strategy (as E. coli infections are risky for neonates of any gestation), extremely preterm newborns less than 28 weeks gestation (which are the ones at highest risk for severe E. coli infections) would have only limited benefit from these protective maternal antibodies. The pups in these preclinical studies were delivered after their full length of pregnancy, which enabled them to fully benefit from the transplacental maternal antibody transfer. While these studies constitute a great proof of concept that maternal immunization against this attenuated E. coli strain can protect the newborn from severe E. coli infections, they do not prove that preterm mice delivered early would be adequately protected from severe E. coli infections. Follow-up studies on larger preterm-delivered mammals, which are beyond the scope of this current manuscript, may be able to prove this concept in future. The authors may nevertheless consider to discuss this limitation of their study.

3) Possibly because of the addition of experiments, some of the figure legends in the text do not seem to match the labeling of the figure panels, which the authors may please correct.

Page 7 lines 150-160: “The IgG2 level was the highest in the serum tested (Figure 2D).” This should be figure 2E instead.

Page 12 lines 270-272: Supplementary figure 3G-H and Supplementary figure 4J and 4K show only E. coli K1 strains, not non-K1 strains.

Page 26 lines 617-618: Figure 2D legend: “Kinetics of antibody titers after immunization with 3 doses by SC.” This figure shows immunization by SC and IP route.

Figure 2D: The abbreviations for subcutaneous and intraperitoneal routes seem reversed.

Figure 3 legend: “A trend of increased regulatory T was observed in the vaccinated group of mice but was significantly different between the two groups ($P > 0.05$).” Please change to ‘...was not significantly different...’

Figure 4 legend page 27 line 645: 10^8 CFU of E. coli K1 S88 should be Figure 4F, 10^7 CFU of E. coli K1 RS218 should be Figure 4G and 10^8 CFU of E. coli K1 RS218 should be Figure 4H.

Figure 5 legend page 28: Only 7 days old mice, not 3 days old mice are shown in Figure 5. Figure panels 5C and 5D are described in the figure legend but the corresponding figure panels are missing.

Figure legend 6 page 28 lines 665-666: “Protection was assessed for 35 days.” This statement only pertains to the 7 days old pups, not to the 3 days old pups.

Supplementary figure 2 legend page 30 line 697: The figure panel H shows pups from immunized mothers.

Supplementary figure 3 legend page 30 line 707: E. coli K1 S88 strain is shown in panel C, not in panel B.

Supplementary figure 4 legend page 31 line 725: The gavage infection route is shown in figure panels J and K, not in G and H.

Supplementary figure 4 legend page 31 line 726-727: The legend states that “Protection was assessed ... for 35 days for 7-day-old pups.” However, the corresponding figure panels only show 21 days on the x-axis.

Reviewer #3 (Remarks to the Author):

Review of main points raised by Rev. 3. I apologise that I do not have the time to re-review the remainder of the manuscript.

1. *aroA* mutant new data. The effort to include new data on the growth in MEM, type I fimbriae expression and global RNAseq is appreciated. The main reason *aroA* is a good mutated background for vaccine strains is that it makes strains auxotrophic for aromatic amino acids and so cannot replicate properly in certain niches in the host. I don't consider your growth curve for WT vs mutant in MEM as 'insignificant', the variation isn't shown so I appreciate that with your repeats it could be argued as such

However, you would expect the *aroA* mutant to have growth issues in more defined media and it would depend exactly on the constituents of the MEM. So, I think you need to acknowledge that you would expect the *aroA* mutant to have growth issues in certain media to be effective as a vaccine. You do show it can replicate in MEM, although slower, but if you control accurately for what you add to your adhesion and invasion assays then this is not as relevant. To be convincing, you could show the actual number of CFU recovered in the assays in Figs 1B/C but you would have to have the numbers at T0 to show the *aroA* mutant did not start at lower levels.....

I certainly think the type 1 fimbrial reduced expression could be important for the adherence and invasion assays but you should caveat that work by stating that type 1 fimbriae expression is controlled by a genetic switch that can be off or on. NUMEC strains probably have altered control of the switch driven by accessory recombinases (*fimB* homologues), so it's possible to take a colony for analysis that could be 'phase on' vs 'phase off'. I think the findings on type 1 fimbriae and *aroA* are interesting but would have to be backed up by more work with *aroA* complementation and analysis of where it feeds into *aim* regulation to be confident it stands up. Appreciate, we are going down a rabbit hole here as it's a bit of a side issue to the main results of the study, I just would like the authors to include some discussion of how type 1 fimbriae are controlled and briefly caveat their findings on T1F expression and growth.

2. As regards the references I provided, I did not really intend you to have to include these specific ones (although it's helpful) just more to point out that you need to acknowledge the breadth of work that underpins your starting point for this work. That many of the different steps are not by themselves novel but that the originality mainly lies with characterising this vaccine strain in the mother-pup models that you have used.

3. The inclusion of the longer survival times for the pups challenged at 7 days is very welcome and I now appreciate the issue with the 3-day old pups. This data is convincing and strengthens the overall manuscript.

4. Line 299 typo *aroA*

REVIEWER COMMENTS Reviewer #1 (Remarks to the Author): “*For obvious reasons, it is difficult to obtain data on the aetiology of neonatal bacterial meningitis and sepsis from patients, so most of our knowledge of these infections has been obtained from animal experimentation using rodents (often rats) or rabbits. The weight of evidence (some of it from human subjects) indicates that the new born infant obtains E. coli K1 from GI tract-colonised mothers; the bacteria then colonise the GI-tract of the infant and subsequently translocate from the child’s gut lumen into the bloodstream via the local lymphatic system before crossing into the cerebrospinal fluid through the choroid plexus and invading the meninges. The bacteria are able to do this because the protective mucous layer in the small intestine has not matured at birth and gives the bacteria the opportunity to interact with the gut epithelium. The best infection models incorporate these aspects.*”

We agree, and infection by gavage with *E. coli* K1 of the new born mice is the main route of infection we used in our previous work that identified AroA as a virulence factor (EBioMedicine. 2023 Feb;88:104439). In this previous work we used both the IP route and the gavage route (that starts with the GI tract colonization described by the reviewer). As the IP route is more severe (Figure S5 from this previous work), we selected this route of infection for the proof of concept efficacy of our approach with the \square aroA vaccine, inferring that a protection against the most severe IP model would mean a de facto infection against the gavage model too. In addition, as previously published, at D7 we can not infect the pups by gavage (the animals are protected in grand part because of the better production of mucus preventing translocation from the GI tract). However, as the reviewer is right and the gavage model is probably closer to the human infection, we have now added protection against different *E. coli* K1 strains using the gavage model at D2-D3 (Three-day-old pups mice born to mothers and immunized either with IP (**Supplementary figure 3G-H**) or SC (**Supplementary figure 4J-K**))

The authors adequately responded to this query by explaining their rational and adding additional data on the gavage route of infection with *E. coli*, which shows that protection is provided in this model as well.

“*Prevention could be effected by eliminating K1 strains from the maternal GI microbiota and this would be good strategy to follow as a couple of studies in pregnant mothers have shown a correlation between neonatal infection and maternal K1 carriage. This might be readily done using K1-specific bacteriophage but surprisingly no one seems to have attempted this. Preventing K1 bacteria from traversing the neonatal gut might conceivably work but would probably require IgA antibody acquisition. Maternal vaccination has been mooted in the past but generally discounted as likely to be ineffective and difficult to achieve in large numbers of at-risk mothers; it would have been useful if the authors had discussed (or even mentioned) this literature.*”
This is now discussed in the manuscript including the strategy using phages to eliminate *E. coli*. (References: 54 and 55).

The authors added these aspects to their discussion section on page 15 lines 33- to 340. References 56 and 57 (not references 54 and 55) comment on the strategy using phages to eliminate *E. coli*, which was correctly cited in the revised manuscript text.

Not mentioned by the authors: IgA antibodies cannot be transferred transplacentally, however can be transferred from the mother to her newborn via colostrum and breastmilk, which then might potentially protect the neonate from intestinal translocation of K1 bacteria.

“The authors use the “unnatural” intraperitoneal route for the introduction of both the vaccine strain and the challenge bacteria to adult mice and conclude that their data provides “proof-of-concept for the development of a live attenuated vaccine against severe E. coli infections in women at risk of preterm delivery” but they offer no strategy for vaccine-derived prevention of infections in humans.

Do they want to prevent infection in mothers or new born or both? How and by what route would the vaccine be administered? How frequently? Without some indication as to how this would be done in clinical practice their concluding statement is meaningless. “

We agree with the reviewer we are now presenting a new set of data using a more translatable route of immunization: the subcutaneous injection. Both mother and new-born mice were still protected with this new vaccination regimen. Results for protection against three different strains of *E. coli* K1 are now presented after SC immunization.

This new regimen is a three-dose subcutaneous vaccination with a one-week interval between doses. We now show:

- (1) the presence of antibodies specific to *E. coli* K1 in immunized female mice (**Figure 2C, D et E**),
- (2) protection of the immunized females against infections with *E. coli* K1 strains (**Supplementary figure 4A, 4B and 4C**)
- (3) protection of pups born from mother immunized SC against infections with *E. coli* K1 strains using two route of infection: IP and gavage (**Supplementary figure 4D, E, F, G, H, I, J, K**).

The authors have adequately responded to this query and added substantial additional data on maternal SC immunization and inoculation of pups with *E. coli* K1 by IP injection and gavage.

“For the last phase of the work they challenge 3- and 7-day-old pups with K1 strains but the route of administration of the challenge is not described in the methods section; I eventually found that they used the intraperitoneal route (probably very difficult to achieve in such small animals) from the legend to figure 6. This needs to be more prominently described in the methods and results sections as there are technical difficulties associated with this procedure in small animals. Figure 6 shows a significant difference in survival of pups from immunized mothers compared to non-immunized but they stop collecting data after 30 hours – what happened after this time point to convince that the difference was due to prevention and not just a delay in the lethal effect?”

The route of infection of the baby mice is now clearly shown in the manuscript, both in the materials and methods section and in the results.

When baby mice are infected at D2-D3, the mother is eating the infected pups, that why we separate the mother from the mice after infection. Then we have to sac the pups at 24-30h to avoid starvation.

However, this issue is not here when the infection is done at D7. Therefore, we can now present the data at much longer times: (35 days post-infection) (**Figure 6D, E, F** for IP) and (**Supplementary figure 4G, H, I** for SC).

New and longer time points have also been added:

-We show new protection data for immunized adults: protection is maintained at 60 days:

Supplemental Figure 3B-D)

-Pups born from immunized mother are still protected against infection at 14days and 28days:

Supplemental Figure 3E-F

We have also added data from the infection of baby D3 mice by gavage (**Supplementary figure 3G and H**) and (**Supplementary figure 4J and 4K**)

- The route of infection is now clearly described. IP injection of 3 days old mice is in my experience very feasible, especially when routinely used in the lab.
- Supplementary figure 4G, H, I show survival data for 7 days old mice followed for 21 days after bacterial challenge, not 35 days (as stated in the figure legend and in the authors' response). This discrepancy needs to be corrected, which I have already pointed out in my own review.
- I concur with the authors that infecting pups during their first few days of life carries a high risk of the newborns being cannibalized by their dams. This is especially true when pups are sick, as is the case in septic mice. In my lab we partially mitigate this problem by only using established female breeders that have successfully raised their newborns for subsequent neonatal sepsis experiments. Despite this measure, when pups become sick from sepsis, a significant proportion of them are cannibalized. The 7 days old mice were now followed for 35 days, which seems appropriate to address this query.

“aroA encodes a gene involved in the biosynthesis of aromatic acids, so it does not directly impact on virulence. The authors found that the mutation reduces both adhesion and invasion so likely affects the expression of a number of virulence factors. It would be useful to have some data on the phenotype of the aroA mutant, in particular with regard to any changes in expression of the main K1 virulence determinant, the polysialic acid capsule.”

As suggested by the reviewers 1 and 3 we further studied the role of *aroA* as a virulence determinant.

-We added the growth conditions used for adhesion and invasion assays (DMEM) (**Figure 1A**) and even studied the impact of the deletion of *aroA* at the full genome scale by performing RNAseq experiments comparing *E. coli* E11WT and *E. coli* Δ *aroA* in LB and DMEM. The main results from these experiments are now presented **Figure 1A, 1D** and **Table supplemental 1**.

A striking result was the impact of the genes for the type 1 fimbriae, a major virulence factor for *E. coli* K1 (**supplemental Figure 5A, B** and **Table supplemental 2**). This surprising result provide a better understanding on the decrease in vitro and in vivo virulence of the Δ *aroA* strain. This result was confirmed by RTqPCR (**Figure 1E**)

The authors substantial new data in response to this query, which adequately addresses the questions.

“The authors imply that protection may be afforded through the induction of antibodies that selectively activate the complement cascade to elicit a humoral bactericidal response, with little cellular immune response. They measure the IgG response in immunized adult mice (Figure 2) but IgM is a much more potent complement activator (which doesn't cross the placental barrier). Also,

two of the four IgG classes are not complement activators and would therefore not contribute greatly discussed. “

As suggested by the reviewer we have extended this part of our study, and have added data evaluating the levels of three IgG isotypes (Ig1, Ig2 and Ig3). We detected a higher level of IgG2 isotype after vaccination. It was previously shown that IgG2 has a high capacity to bind

to a novel gamma chain-dependent IgG activating Fc receptor, FcgammaRIV, and has high protective and pathogenic properties. (Reference 46)

The authors responded adequately to this question. Furthermore, they now determined type I fimbriae a major virulence factor.

“Also, many K1 clinical isolates are complement-resistant and may therefore circumvent protection.”

We added data on the bactericidal activity of the serum with two other strains of *E. coli* K1 (S524 and S395 (**Supplementary figure 3A**)) but mentioned in the text that other strains could be resistant in this assay (with a ref associated).

However, in vivo, the killing of microorganisms by maternal antibodies is not mediated only by complement. We and other have reported reviews summarizing the literature on these mechanisms and this review has also been added to the text (Reference 29).

This question has been adequately addressed by the authors.

Reviewer #1 (Remarks to the Author):

- Supplementary figure 4G, H, I show survival data for 7 days old mice followed for 21 days after bacterial challenge, not 35 days (as stated in the figure legend and in the authors' response). This discrepancy needs to be corrected, which I have already pointed out in my own review. :

This typo has now been fixed.

Reviewer #2 (Remarks to the Author):

As previously stated, this is a highly relevant topic of scientific and clinical relevance, and experiments were well conducted and are supporting the conclusions. The authors have added several additional informative experiments in support of their studies as well as additional references and discussion points, and as such their revised manuscript version is significantly improved and adequately addresses most of the reviewers' critiques.

We thank the reviewer for this positive comment.

Nevertheless, I do have a few remarks.

1) Introduction -page 3 lines 49-50: "With a death rate of 35% for preterm newborns and 18% for preterm children up until the age of 5 years, prematurity is a true public health problem worldwide 2,3."

Per reference 3 (Walani), "... preterm birth is the leading cause of death among children, accounting for 18% of all deaths among children aged under 5 years and as much as 35% of all deaths among newborns (aged <28 days)." This means that 35% of all deaths among newborns are due to prematurity, rather than that 35% of all preterm newborns do not survive. Please correct this statement.

We have corrected this statement by saying: Lines 49-50

"With 35% of all deaths among newborns due to prematurity, prematurity which also accounts for 18% of all deaths among children aged under 5 years is a true public health problem worldwide".

2) I would also like to remark the following: The authors state that "women with a history of preterm delivery have a high risk of recurrence and therefore constitute a target population for the development of a vaccine against neonatal E. coli infections." Undoubtedly preterm neonates, especially very preterm neonates, are at high risk for severe infections such as E. coli sepsis and meningitis. The authors show that immunization of female mice with the attenuated E. coli K1 E11 Δ aroA vaccine leads to transfer of protective IgG antibodies to the pups. Transplacental IgG transfer to the human fetus takes primarily place during the last trimester of pregnancy, reaching only about 50% by 28 to 32 weeks gestation (e.g., PMID: 22235228). While late preterm and term newborns could potentially greatly benefit from this strategy (as E. coli infections are risky for neonates of any gestation), extremely preterm newborns less than 28 weeks gestation (which are the ones at highest

risk for severe E. coli infections) would have only limited benefit from these protective maternal antibodies. The pups in these preclinical studies were delivered after their full length of pregnancy, which enabled them to fully benefit from the transplacental maternal antibody transfer. While these studies constitute a great proof of concept that maternal immunization against this attenuated E. coli strain can protect the newborn from severe E. coli infections, they do not prove that preterm mice delivered early would be adequately protected from severe E. coli infections.

Follow-up studies on larger preterm-delivered mammals, which are beyond the scope of this current manuscript, may be able to prove this concept in future. The authors may nevertheless consider to discuss this limitation of their study.

As suggested by the reviewer we have now added a paragraph to discuss this limitation of our study.

“A limitation of our study is related to the transplacental transfer of IgG to the human foetus that occurs mainly during the last trimester of pregnancy and only reaches around 50% between 28- and 32-weeks’ gestation. Therefore, while late preterm and at term neonates could potentially benefit greatly from the strategy described in this work, extremely preterm neonates born after less than 28 weeks gestation would probably benefit only to a limited extent from these protective maternal antibodies.

Another limitation of our study is the lack of evidence that prematurely born mice would be adequately be protected against severe E. coli infections. Further follow-up studies in larger preterm mammals, beyond the scope of this current work, may be able to prove this concept in the future.

However, despite these limitations, our study provides an excellent preclinical proof of concept that maternal immunisation with our attenuated live vaccine can protect neonates against severe E. coli infections.”

3) Possibly because of the addition of experiments, some of the figure legends in the text do not seem to match the labeling of the figure panels, which the authors may please correct.

This has now been fixed.

Page 7 lines 150-160: “The IgG2 level was the highest in the serum tested (Figure 2D).” This should be figure 2E instead.

We have corrected this.

Page 12 lines 270-272: Supplementary figure 3G-H and Supplementary figure 4J and 4K show only E. coli K1 strains, not non-K1 strains.

We have corrected this: E. coli K1

Page 26 lines 617-618: Figure 2D legend: “Kinetics of antibody titers after immunization with 3 doses by SC.” This figure shows immunization by SC and IP route.

We have corrected this.

Figure 2 D shows the antibody titers after immunization with 3 doses (IP route on top, SC route in the bottom).

Figure 2D: The abbreviations for subcutaneous and intraperitoneal routes seem reversed.

We have corrected this.

Figure 3 legend: "A trend of increased regulatory T was observed in the vaccinated group of mice but was significantly different between the two groups ($P > 0.05$)." Please change to "...was not significantly different..."

We have corrected this.

Figure 4 legend page 27 line 645: 10^8 CFU of E. coli K1 S88 should be Figure 4F, 10^7 CFU of E. coli K1 RS218 should be Figure 4G and 10^8 CFU of E. coli K1 RS218 should be Figure 4H.

We have corrected this.

Figure 5 legend page 28: Only 7 days old mice, not 3 days old mice are shown in Figure 5. Figure panels 5C and 5D are described in the figure legend but the corresponding figure panels are missing.

We have corrected this:

(A): Seven days after the last vaccination dose, immunized and non-immunized BALB/c female mice were mated with BALB/c males. Newborn mice born to immunized and non-immunized mothers were sacrificed at Day 7 (n=4) and their sera collected for (B) evaluation of the presence of maternal antibodies titers specific to *E. coli* K1 E11 WT, S88 and RS218 by whole bacterial cell ELISA.

Figure legend 6 page 28 lines 665-666: "Protection was assessed for 35 days." This statement only pertains to the 7 days old pups, not to the 3 days old pups.

We have corrected this.

Supplementary figure 2 legend page 30 line 697: The figure panel H shows pups from immunized mothers.

We have corrected this.

Supplementary figure 3 legend page 30 line 707: E. coli K1 S88 strain is shown in panel C, not in panel B.

We have corrected this.

Supplementary figure 4 legend page 31 line 725: The gavage infection route is shown in figure panels J and K, not in G and H.

We have corrected this.

Supplementary figure 4 legend page 31 line 726-727: The legend states that “Protection was assessed ... for 35 days for 7-day-old pups.” However, the corresponding figure panels only show 21 days on the x-axis.

We have corrected this.

Reviewer #3 (Remarks to the Author):

Review of main points raised by Rev. 3. I apologise that I do not have the time to re-review the remainder of the manuscript.

1. *aroA* mutant new data. The effort to include new data on the growth in MEM, type I fimbriae expression and global RNAseq is appreciated

We thank the reviewer for this positive comment.

*The main reason *aroA* is a good mutated background for vaccine strains is that it makes strains auxotrophic for aromatic amino acids and so cannot replicate properly in certain niches in the host.*

We agree with the reviewer a comment and a sentence has been added in the introduction

I don't consider your growth curve for WT vs mutant in MEM as 'insignificant', the variation isn't shown so I appreciate that with your repeats it could be argued as such

We agree that the wording was confusing. Our purpose was to show that additional mechanistic explanations would provide a better understanding of the adhesion assays results. But the growth defect certainly also played a role. As suggested by the reviewer, we have now added the CFU counts as T0 and adhesion, invasion. The numerical CFU values for adhesion and invasion are shown in Supplementary Figure 1C, D and F)

*However, you would expect the *aroA* mutant to have growth issues in more defined media and it would depend exactly on the constituents of the MEM.*

We agree with the reviewer that in some media, the *aroA* mutant has a strong defect:

Bow a growth defect found in a minimum medium that is countered by the addition of aromatic Amino Acid. We carried out an auxotrophicity test for amino acids in the minimal M9 medium. We

observed a difference in growth between *E. coli* K1 E11 WT and Δ aroA in M9 minimal medium not supplemented with amino acid. The mutant did not grow in minimal medium whereas it grew in minimal medium supplemented with amino acid. These results has been added as supplementary figure 1B.

So, I think you need to acknowledge that you would expect the aroA mutant to have growth issues in certain media to be effective as a vaccine.

We agree with the reviewer and as stated above, added a sentence about this characteristic in the introduction Lines 82- 84 and added the Supplemental Figure 1B.

You do show in can replicate in MEM, although slower, but if you control accurately for what you add to your adhesion and invasion assays then this is not as relevant. To be convincing, you could show the actual number of CFU recovered in the assays in Figs 1B/C but you would have to have the numbers at T0 to show the aroA mutant did not start at lower levels

As suggested by the reviewer we have compared the CFU at T0 and showed that the AroA mutant did not start at a lower level: (Supplementary figure 1C).

This sentence has been added in the main text:

“At T0 the CFU level of the WT and the aroA mutants strains were 3,90E+05 and 4,60E+05, confirming that the aroA mutant did not start at a lower level (Supplemental Figure 1C-E).” Lines 117-119.

.....

I certainly think the type 1 fimbrial reduced expression could be important for the adherence and invasion assays but you should caveat that work by stating that type 1 fimbriae expression is controlled by a genetic switch that can be off or on.

As suggested by the reviewer, a sentence has been added this comment in the discussion:

“The role of the fimbriae probably play an important role in the results found in the adherence and invasion assays, however, we also have to take into account that type 1 fimbriae expression is controlled by a genetic switch that can easily been turned off or on”. PMID : 10601203. Lines 310-313.

NUMEC strains probably have altered control of the switch driven by accessory recombinases (fimB homologues), so its possible to take a colony for analysis that could be ‘phase on’ vs ‘phase off’. I think the findings on type 1 fimbriae and aroA are interesting but would have to be backed up by more work with aroA complementation and analysis of where it feeds into aim regulation to be confident it stands up. Appreciate, we are going down a a rabbit hole here as its a bit of a side issue to the main results of the study, I just would like the authors to include some discussion of how type 1 fimbriae are controlled and briefly caveat their findings on T1F expression and growth.

As stated above, a new paragraph has been added in the discussion Line 310-313.

2. As regards the references I provided, I did not really intend you to have to include these specific ones (although its helpful) just more to point out that you need to acknowledge the breadth of work that underpins your starting point for this work. That many of the different steps are not by themselves novel but that the originality mainly lies with characterising this vaccine strain in the mother-pup models that you have used.

We thank the reviewer for his positive comments.

3. The inclusion of the longer survival times for the pups challenged at 7 days is very welcome and I now appreciate the issue with the 3-day old pups. This data is convincing and strengthens the overall manuscript.

We thank the reviewer for this positive comment

4. Line 299 typo aroA

We have corrected this.